# PET-CT in Clinical Adult Oncology—IV. Gynecologic and Genitourinary Malignancies

**DOI:** 10.3390/cancers14123000

**Published:** 2022-06-18

**Authors:** Ahmed Ebada Salem, Gabriel C. Fine, Matthew F. Covington, Bhasker R. Koppula, Richard H. Wiggins, John M. Hoffman, Kathryn A. Morton

**Affiliations:** 1Department of Radiology and Imaging Sciences, University of Utah, Salt Lake City, UT 84132, USA; ahmed.salem@utah.edu (A.E.S.); gabriel.fine@hsc.utah.edu (G.C.F.); matthew.covington@hsc.utah.edu (M.F.C.); bhasker.koppula@hsc.utah.edu (B.R.K.); richard.wiggins@hsc.utah.edu (R.H.W.); john.hoffman@hci.utah.edu (J.M.H.); 2Department of Radiodiagnosis and Intervention, Faculty of Medicine, Alexandria University, Alexandria 21526, Egypt; 3Intermountain Healthcare Hospitals, Summit Physician Specialists, Murray, UT 84123, USA

**Keywords:** [^18^F]-fluorodeoxyglucose, PET, prostate specific membrane antigen (PSMA), fluciclovine, gynecologic malignancy, prostate cancer, renal cell carcinoma, urothelial carcinoma, testicular cancer, penile cancer

## Abstract

**Simple Summary:**

Positron emission tomography (PET), typically combined with computed tomography (CT), has become a critical advanced imaging technique in oncology. With concurrently acquired positron emission tomography and computed tomography (PET-CT), a radioactive molecule (radiotracer) is injected in the bloodstream and localizes to sites of tumor because of specific cellular features of the tumor that accumulate the targeting radiotracer. The CT scan provides information to allow better visualization of radioactivity from deep or dense structures and to provide detailed anatomic information. PET-CT has a variety of applications in oncology, including staging, therapeutic response assessment, restaging and surveillance. This series of six review articles provides an overview of the value, applications, and imaging interpretive strategies for PET-CT in the more common adult malignancies. The fourth report in this series provides a review of PET-CT imaging in gynecologic and genitourinary malignancies.

**Abstract:**

Concurrently acquired positron emission tomography and computed tomography (PET-CT) is an advanced imaging modality with diverse oncologic applications, including staging, therapeutic assessment, restaging and longitudinal surveillance. This series of six review articles focuses on providing practical information to providers and imaging professionals regarding the best use and interpretative strategies of PET-CT for oncologic indications in adult patients. In this fourth article of the series, the more common gynecological and adult genitourinary malignancies encountered in clinical practice are addressed, with an emphasis on Food and Drug Administration (FDA)-approved and clinically available radiopharmaceuticals. The advent of new FDA-approved radiopharmaceuticals for prostate cancer imaging has revolutionized PET-CT imaging in this important disease, and these are addressed in this report. However, [^18^F]F-fluoro-2-deoxy-d-glucose (FDG) remains the mainstay for PET-CT imaging of gynecologic and many other genitourinary malignancies. This information will serve as a guide for the appropriate role of PET-CT in the clinical management of gynecologic and genitourinary cancer patients for health care professionals caring for adult cancer patients. It also addresses the nuances and provides guidance in the accurate interpretation of FDG PET-CT in gynecological and genitourinary malignancies for imaging providers, including radiologists, nuclear medicine physicians and their trainees.

## 1. Introduction

Hybrid positron emission tomography–computed tomography (PET-CT) is an invaluable advanced diagnostic imaging modality in oncology with a variety of applications, including initial staging of cancer, assessment of response to therapy, restaging and longitudinal surveillance for recurrence. Historically, clinical oncologic PET utilized [^18^F]F-fluoro-2-deoxy-d-glucose (FDG) almost exclusively. This remains a mainstay for PET imaging of gynecologic and many genitourinary malignancies. However, there are three new FDA-approved agents for prostate cancer imaging that are now available, Axumin^®^ ([^18^F]-anti-1-amino-3-[^18^F]-fluorocyclobutane-1-carboxylic acid, also known as FACBC, Blue Earth Diagnostics, Oxford, UK) and two agents that target prostate specific membrane antigen, PYLARIFY^®^ ([^18^F]-piflufolastat, also called DCFPyL or Pyl, Lanheus/Progenics, Millerica, MA, USA) and LOCAMETZ^®^ ([^68^Ga]-PSMA-11, PSMA-HBED-CC, gozetotide, Advanced Accelerator Applications, Millburn, NJ, USA). The availability of a greater diversity of radiopharmaceuticals, technological improvements in instrumentation such as digital PET-CT, and acquired global clinical and imaging experience have placed PET-CT at the forefront of gynecologic and urologic imaging.

The goal of this six-part series of articles is to provide a compendium of cohesive practical information regarding the best use of PET-CT for some of the most common adult malignancies. This includes key imaging questions that can be effectively addressed by PET-CT imaging, as well as limitations, pitfalls and nuances that should be considered in the accurate interpretation of PET-CT scans for these indications. The current fourth article in the six-part series addresses gynecologic (GYN) and genitourinary (GU) malignancies. A discussion of the role of FDG PET-CT for all of the many GYN and GU tumors in this category is beyond the scope of a single review article. Rather, this review focuses on the most common adult GYN and GU malignancies that may be encountered in clinical practice.

The target readers for this review are clinicians caring for cancer patients as well as imaging providers (radiologists and nuclear medicine practitioners) and their trainees. As such, the focus of this review article is also on US Food and Drug Administration (FDA)-approved and clinically accessible radiopharmaceuticals and methods, rather than research tracers, investigational analytical methods or tracers that require a local cyclotron for production. Nonetheless, brief reference is made, where appropriate, to some of the most promising radiopharmaceuticals currently under development.

Other prominent topics in imaging research that are of great interest today are also not addressed in depth because they do not directly apply to clinical practice today. For example, it is also acknowledged that there is a large body of literature relating PET imaging characteristics to prognosis and outcome for specific cancers. Although significant, this is not the focus of this review because PET imaging characteristic predictors of prognosis and outcome, other than those related to treatment-response, are typically not included in consensus and evidence-of-medicine based treatment algorithms. Additionally, an enormous body of research has been devoted to the development of artificial intelligence (AI) and machine learning to develop radiomic profiling of tumors based on combinations of imaging features. While of significant interest, these radiomic applications have not yet found their way into routine clinical practice and are dealt with only briefly in this review. Finally, because hybrid PET-magnetic resonance imaging (PET-MRI) scanners are in limited international use, the focus herein is on PET-CT because if its widespread availability. However, basic principles described are also applicable to PET-MRI.

## 2. Gynecological Malignancies

### 2.1. Normal Physiologic and Benign Patterns

FDG PET-CT is an important imaging tool for management of patients with gynecological malignancies [1,2,3]. In this regard, it has been shown to be of great value not only in characterizing the primary tumor, but also in detection of intrapelvic extension of the tumor, involvement of pelvic and retroperitoneal nodes and in identifying distant metastases [4,5]. In addition, there is evidence that FDG PET-CT is useful in assessing treatment response [6]. However, it is important that readers be aware of a number of physiologic and benign patterns that, if not recognized, create potential pitfalls in accurate interpretation of findings [7,8]. Magnetic resonance imaging (MRI) remains a critical modality in assessing spread of disease within the pelvis and, in this regard, FDG PET-CT is an adjunct to conventional imaging in gynecologic malignancies.

Due to the heterogeneous textural patterns on contrast-enhanced CT, MRI and FDG PET-CT in many gynecological malignancies, these tumors represent opportunities for radiomic development using characteristics of multiple imaging modalities for prognostic applications and tumor phenotyping. In primary cervical cancer, extracted radiomic features have been shown to differentiate between local and metastatic cervical cancer [9]. This strategy has been applied in advanced, high grade serous ovarian cancer, where radiomic features derived from FDG PET-CT have been applied to predict progression free survival [10]. FDG PET-CT has also been used for prognostic information, including disease progression and mortality, in ovarian carcinoma [11].

The mainstay in PET-CT imaging of gynecological malignancies is FDG. However, the development of novel radiopharmaceuticals is a constantly evolving field. Notable is the development of radiolabeled fibroblast activation inhibitor protein, [^68^Ga]-FAPI, which has been used in patients with a number of types of gynecologic malignancies [12].

It should be noted that a low-dose, non-contrast enhanced CT, when performed in conjunction with an FDG PET-CT scan, is significantly limited when compared to a PET-CT scan performed with diagnostic exposure and both intravenous and oral contrast. On separately performed exams, changes in bowel contents and peristalsis between the PET and CT limit precise co-registration. For this reason, it is our recommendation that oral and iodinated contrast be administered with a diagnostic dose CT when performing an FDG PET-CT for gynecologic malignancy. Even if a low dose CT is utilized, the administration of iodinated intravenous and oral contrast significantly enhances CT image quality.

There are a number of normal physiologic patterns that are important to recognize in evaluating the female pelvis by FDG PET-CT. The endometrium typically is diffusely metabolically active in premenopausal women during the first 4 days of the menstrual cycle, as well as during the ovulatory phase. Concomitantly, sites of endometriosis may show increased uptake during those same times as well (Figure 1). Increased endometrial metabolic activity in post-menopausal women is more worrisome, and may be associated with hyperplasia or even malignancy and should be investigated. Benign leiomyomas (fibroids) may differ widely in degree of hypermetabolism but are more typically low in activity [13] (Figure 2). The post-partum uterus is typically large and the endometrium is thickened and metabolically active (Figure 3). During the ovulatory phase/mid cycle, increased activity is often observed in one or both fallopian tubes (Figure 4) [14]. Corpus luteum cysts or maturing ovarian follicles typically are hypermetabolic on FDG PET-CT, with corpus luteum cysts showing a typical circular pattern on contrast enhanced CT (Figure 5).

There are a number of additional sources of possible false positives on FDG PET-CT in the female pelvis. Pelvic inflammatory disease can result in hypermetabolic changes (Figure 6). Urine contamination on the vulva or refluxed into the vagina is a common finding. A fistula, whether or not it connects to the urinary bladder, will also cause increased uptake (Figure 7). Bladder diverticula and ureters can be confused with pelvic disease if not carefully examined on CT. Hypermetabolic inflammatory changes, particularly produced by radiation, will often persist for up to 2–3 months post treatment. Diverticulitis can be confused with a peritoneal implant, and high-quality comparative CT is critical in making this distinction.

False negative FDG PET-CT scans in gynecologic malignancy can be seen in a number of situations, for example, with necrotic, mucinous, low grade tumors, tiny lesions and some sarcomas [15]. High physiologic or pharmacologic (e.g., metformin-induced) bowel activity may obscure serosal implants. Finally, high urinary bladder activity may result in an apparent decrease in activity across the remainder of the pelvis, hiding areas of metabolically active tumor. High serum glucose levels can decrease uptake of FDG in sites of tumor. Knowledge of the many pitfalls and challenges of FDG PET-CT is critical for accurate imaging of gynecologic malignancies.

### 2.2. Ovarian Epithelial, Fallopian Tube and Primary Peritoneal Carcinoma

Ovarian cancer is the eighth most common cancer in women and the most lethal gynecological malignancy [16]. It affects 1 in 71 women in the US. The five-year survival rate of ovarian cancer is 48.6%, although prognosis depends upon the stage of disease, ranging from 87% for Stage 1A to 11% for Stage IV [17]. Ovarian epithelial tumors comprise 95% of ovarian malignancies, with germ cell and sex-cord stromal tumors representing a small percentage. Ovarian epithelial tumors consist of a number of subtypes, including mucinous. However, the large majority are high grade serous tumors [18]. High grade serious ovarian cancer, fallopian tube carcinoma and primary peritoneal carcinoma are considered as a single entity by the NCCN and common practice, as their clinical features, risk factors and management are essentially the same.

Epithelial ovarian carcinoma are often found incidentally, with either pelvic pain or imaging performed for an unrelated reason, or by surgery performed for other indications [19]. Adnexal cysts can either be functional or neoplastic (benign or malignant), and features that distinguish between these categories depend upon the age of the patient, sonographic or MRI complexity, the size of the cyst and serum CA-125 levels [20]. Screening to identify early disease or prophylactic oophorectomy is recommended in certain high-risk populations, such as patients with BRCA1/2, who are at a higher risk of developing ovarian, fallopian tube and primary peritoneal carcinoma [21]. However, epithelial ovarian cancer typically presents late in most patients, with ascites, abdominal distension, bowel obstruction and intraabdominal deep venous thrombosis. The diagnosis of ovarian cancer is typically made by cytology of ascitic fluid or at surgical removal of suspicious adnexal masses or cysts. Care is taken to avoid spillage of cystic contents intraperitoneally upon removal of the mass, particularly with laparoscopic approaches. To do so may seed the peritoneum with tumor (Figure 8), create pseudomyxomatous peritoneii from mucinous tumors or convert a localized ovarian cancer or cyst of low-grade malignancy to diffuse peritoneal spread. Treatment of ovarian cancer is typically by surgery, with either curative intent or for aggressive debulking or cytoreduction. Cytotoxic chemotherapy is typically given to patients either before or after surgery. In BRCA mutation patients, PARP and VEGF inhibitor therapy may be of additional benefit [22].

Initial imaging evaluation of an adnexal mass or cyst is by ultrasound or MRI. Initial imaging of suspected ovarian cancer relies upon contrast enhanced CT and MRI. FDG PET-CT has not been found to play a role in establishing the initial diagnosis of ovarian cancer or establishing whether an adnexal mass or cyst is malignant or benign. However, PET-CT typically shows hypermetabolism in the solid components of ovarian cancer (Figure 9) and may be used to identify metastases when advanced disease is suspected. Many false positives are identified on PET-CT in adnexal masses, including endometriomas, dermoid cysts and pelvic inflammatory disease and developing follicles. False negatives include mucinous and low-grade malignancies and small tumors.

The NCCN supports the use of FDG PET-CT for monitoring response to chemotherapy, for surveillance and for evaluating recurrent disease [23]. In this regard, PET-CT can be performed instead of contrast enhanced CT or MRI. The imaging evaluation is important prior to cytoreductive surgery. There was no real advantage of one modality over the other when PET-CT, MRI or CT were compared in the preoperative assessment of peritoneal cancer index (a quantitative assessment of the extent of peritoneal spread), although others have reported a slight advantage of PET-CT and MRI over CT alone [24,25]. However, FDG PET-CT does offer a more compelling advantage in identifying extraperitoneal metastatic disease [26]. It is not uncommon for metastases to occur to the ovary, such as with breast cancer, gastric cancer (Krukenberg tumor, Figure 10), mucinous pancreatic cancer, neuroendocrine tumors or with mucinous colon cancer (Figure 11). In this regard, FDG PET-CT may play a role in identifying the site of the primary tumor.

### 2.3. Uterine Neoplasms: Endometrial Carcinoma, Uterine Sarcoma and Leiomyoma

#### 2.3.1. Endometrial Carcinoma

Uterine cancer consists of two categories: malignant epithelial (endometrial carcinoma) and malignant mesenchymal (uterine sarcomas). Endometrial carcinoma (corpus uterine cancer) is the most common gynecological malignancy, with a lifetime occurrence of 2.8% [27]. Most are adenocarcinomas, although some may have squamous components. Higher risk histologic subtypes include serous, clear cell, undifferentiated or carcinosarcomas. If found early, the prognosis is favorable, with an overall survival rate of >80%, although survival rate depends upon tumor stage. Endometrial carcinoma spreads to regional, then distant, lymphatics as well as through the fallopian tubes to result in peritoneal carcinomatosis, and by hematogenous spread to distant metastatic sites.

Seventy-five percent of women with endometrial carcinoma are post-menopausal and present with vaginal bleeding. Pre- or peri-menopausal women may present with worsening menstrual bleeding or intermenstrual bleeding, but are more difficult to diagnose, clinically. Risk factors include unopposed estrogen replacement, obesity, possibly tamoxifen and hereditary nonpolyposis colon cancer (HNPCC) [28]. Diagnosis is typically made by biopsy, dilation and curettage (D&C) or hysterectomy. Treatment is by surgery, followed by chemotherapy +/− radiation therapy [29]. For locally advanced disease or recurrent disease, cytoreductive surgery improves outcome [30]. Pelvic and periaortic (below the renal vessels) lymphadenectomy is advocated for early high-risk disease or initial locally advanced disease [31]. Treatment of advanced or refractory disease typically includes chemotherapy, hormonal therapy, tamoxifen, aromatase inhibitors and targeted agents, such as mTOR inhibitors [32,33].

Identification and characterization of the primary endometrial carcinoma typically first involves transvaginal ultrasound. However, the initial staging of endometrial carcinoma is established by FIGO criteria, which depends largely on the physical exam and chest X-ray. MRI is typically used for clarification of the extent of pelvic disease, with contrast enhanced CT typically employed initially for staging. Imaging methods have all been reported to be less sensitive than surgical staging of nodal metastases, although FDG PET-CT offers greater sensitivity (80%) than either nodal size (65%), nodal morphology (75%) or diffusion-weighted imaging (DWI)-MRI (70%) for detection of nodal involvement [34]. FDG PET-CT is a superior modality for identifying nodal disease preoperatively and recurrent disease postoperatively, with an overall accuracy of 88% for identifying nodal disease and 93% for recurrence [35]. The National Comprehensive Cancer Network (NCCN) considers the use of FDG PET-CT to be appropriate for initial staging if metastatic endometrial carcinoma is suspected and for restaging of recurrence [36]. The Society of Gynecologic Oncology (SGO) and European Society of Medical Oncology (ESMO) support PET-CT over CT for cases where recurrence is likely [32,37]. Since endometrial carcinoma is a relatively common condition, it is not only important to recognize the entity and its manifestations on FDG PET-CT performed for patients with known endometrial carcinoma but also for patients scanned for other reasons, in whom the opportunity for early recognition of unsuspected endometrial carcinoma might be achieved. Hypermetabolic endometrial thickening, while often physiologic in pre-menopausal women, is not normal in a post-menopausal woman and typically indicates hyperplasia, at a minimum. By FDG PET-CT, uterine carcinomas are typically low in CT attenuation, but markedly hypermetabolic on PET (Figure 12). In women on tamoxifen for breast cancer, attention to the uterus on FDG PET-CT is important because tamoxifen increases risk of uterine hyperplasia, polyps, uterine carcinoma and uterine sarcoma. By FDG PET-CT, both polyps and uterine carcinoma are similarly hypermetabolic and may be associated with increased fluid in the endometrial cavity (Figure 13).

#### 2.3.2. Uterine Sarcoma

Uterine sarcomas comprise only 3% of uterine tumors [38]. There are a number of types of uterine sarcomas that differ significantly in aggressive behavior. Due to the rare and diverse nature of the tumors, the literature, which consists mostly of small series and case reports, is sparse in evidence of medicine to support a specific role of FDG PET-CT in managing uterine sarcomas. However, limited evidence supports that FDG PET-CT is likely useful, as it is in other gynecological malignancies, in the management of uterine sarcomas [39]. Of note, carcinosarcoma is now considered an aggressive form of endometrial carcinoma and is managed accordingly [36].

Leiomyosarcoma is the most common uterine sarcoma, comprising 1% of uterine tumors. A diagnostic challenge in managing patients is to distinguish between uterine leiomyomas (fibroids) and leiomyosarcoma. A very large or rapidly growing uterine mass will often trigger an assumption that it is a leiomyosarcoma. Since leiomyomas are common and leiomyosarcomas are rare, the risk is that many unnecessary surgeries will be performed for the indication of possible leiomyosarcoma. Leiomyosarcomas of the uterus are strongly FDG avid. As mentioned above, uterine leiomyomas can differ greatly in magnitude of hypermetabolism on FDG PET-CT, and this can differ with phase of the menstrual cycle. Some leiomyomas are metabolically active for unknown reasons (Figure 2, above). An additional complexity is that benign uterine leiomyomas may occasionally metastasize, such as to the lungs. The value of FDG PET-CT in assessing the malignant leiomyosarcomas vs. benign metastasizing uterine leiomyomas has not been clearly established. However, available evidence supports that magnitude of uptake on FDG PET is variable in the benign uterine leiomyomas as well as benign metastasizing leiomyomas, regardless of cellular features [40,41]. Therefore, based on this information, it is unlikely that FDG PET-CT can reliably distinguish between benign leiomyomas and leiomyosarcomas.

Low grade uterine stromal sarcomas are rare tumors that have a propensity to recur with metastatic disease many years after initial identification and presumed successful definitive treatment of the primary tumor. Despite the low-grade features of these tumors, they tend to be strongly FDG-avid [42]. When these tumors recur, they tend to be within the pelvis or lungs, and are also hypermetabolic. High grade stromal sarcomas are aggressive tumors with a poor prognosis [43]. However, the distinction between low and high grade uterine stromal sarcomas requires a histological evaluation of features at the tumor-myometrial interface (Figure 14). Neither MRI nor FDG PET is thought to be useful in making this distinction [44].

Another low-grade uterine sarcoma is adenosarcoma without sarcomatous overgrowth. The role of FDG PET in this tumor type is not established. This entity as well as low-grade endometrial stromal tumor are often managed with hormonal management, either pharmacological or by bilateral salpingo-oophorectomy (BSO), with radiation reserved for higher stage tumors. Other rare uterine sarcomas, such as adenosarcoma, adenosarcoma with sarcomatous overgrowth and perivascular endothelial cell tumor are similar in that there are no specific data regarding the role of FDG PET in these subtypes of sarcomas. Available limited evidence with all sarcomas considered as a single group suggests that FDG PET-CT may have a low sensitivity in detection of nodal disease. The best role of FDG PET-CT is in the evaluation of recurrence of uterine sarcoma, where pooled sensitivity and specificity is >90% [45].

### 2.4. Cervical Cancer

Although relatively uncommon in the US (the 20th most common cancer), cervical cancer is the 3rd most common malignancy and the highest cause of cancer deaths among women in the world [46]. The cause of cervical cancer is human papilloma virus (HPV), with multiple subtypes being associated with risks of varying degrees. With the advent of the nine-valent HPV vaccine and aggressive screening by PAP tests, the incidence in the US has declined. Seventy-five percent of cervical cancers are squamous cell, with the remainder predominantly represented by adenocarcinoma and adenosquamous cell types [47].

Cervical carcinoma in situ (Stage 0) can be treated with localized ablation. For Stage I disease, the size and depth of disease must be established histologically. For Stage II disease confined to the cervix, the size of the tumor determines substage (IB1 < 2 cm, IB2 > 2 cm to <4 cm, IB3 (>4 cm) (Figure 15). Disease that extends into the upper third of the vaginal region only is Stage IIA (Figure 16). That which involves parametrial regions without reaching the pelvic sidewall is Stage IIB. Stage III disease consists of spread to the lower third of the vagina only (IIIA) (Figure 17), pelvic sidewall or is associated with hydronephrosis (IIIB) (Figure 18), or involves pelvic (IIIC1) or periaortic (IIIC2) lymph nodes. Stage IV disease consists of spread to the rectum or bladder (IVA) or distant sites (IVB) [48].

For Stage II and higher cervical cancer, as well as recurrent disease, the treatment varies with stage, and accurate imaging is critical for appropriate management. The treatment of higher stage tumors can include surgery, brachytherapy, radiation therapy and chemotherapy. Fertility sparing vaginal trachelectomy is often employed for fertility preservation in early stage disease. The extent of parametrial spread in stage II and III disease typically relies upon multimodality MRI. However, the value of FDG PET-CT in the staging and disease assessment of Stage I disease and higher has been well-established [49]. NCCN recommends FDG PET-CT as the preferred imaging modality for the initial work-up of Stage I and higher disease. For detection of regional and periaortic positive lymph nodes, FDG PET-CT offers a sensitivity of 88% and a specificity of 93% [50]. In assessing response to treatment, NCCN endorses FDG PET-CT as optional for Stage I disease and the preferred imaging method for stages II and higher, preferably performed at 3–6 months post completion of treatment [51]. Complete metabolic response is a significant prognostic indicator of treatment success [52]. An FDG PET scan performed 3 months following treatment predicts a three-year progression free survival of 78% with complete metabolic response, 33% with partial metabolic response and 0% with obvious progressive disease [53]. With suspected recurrence or metastatic disease, PET-CT is also the preferred imaging modality, with optional MRI of the pelvis. Post-treatment persistence of disease on FDG PET-CT may be used to justify more aggressive additional treatment, such as adjuvant treatment or surgical pelvic exenteration [54]. The presence of metabolic heterogeneity by the tumor has been proposed as an indication for possible dose escalation in radiation treatment but is a parameter that has not yet reached common use [55].

### 2.5. Vaginal Cancer

Primary vaginal cancer is rare, comprising less than 1% of all gynecological malignancies. Most cases are squamous cell carcinoma arising from other structures, such as the vulva or cervix, or by invasion from other pelvic malignancies. In those cases, vaginal cancer is approached according to the guidelines of those other malignancies. Other tumor types, such as melanoma, adenocarcinoma and metastases, also occur less frequently. Most cases of primary vaginal squamous cell carcinoma are mediated by several HPV subtypes, although intrauterine exposure to diethylstilbestrol (DES), older age, smoking history and history of multiple sexual partners have also been associated with vaginal cancer.

There are limited data on the use of FDG PET-CT in vaginal cancer, but available evidence supports that FDG PET-CT detects more primary tumors and involved nodes than does CT [56,57]. The NCCN does not have guidelines for this specific subcategory of malignancy, and FIGO recommendations are somewhat archaic for imaging (chest and skeleton X-rays). However, the American College of Radiology’s appropriate use criteria (AUC) for primary vaginal cancer support that the use of FDG PET-CT is “usually appropriate” for initial staging, post-treatment assessment and assessment of suspected recurrence [58]. Despite limited data, FDG PET-CT is considered of value in the management of patients with primary vaginal carcinoma (Figure 19) and in the spread of adjacent primary tumor (particularly cervical carcinoma) to the vagina (Figure 16 and Figure 17).

### 2.6. Vulvar Carcinoma

Vulvar carcinoma accounts for 5% of gynecological malignancies and can be seen in younger women, with risk factors including early age or high-risk sexual activity and HPV infection [59]. In this population, the disease tends to be associated with diffuse vulvar intraepithelial neoplasia (VIN). However, most cases occur in older women without clear antecedent risk factors [60]. Most cases of vulvar cancer are squamous cell carcinoma. Primary mucosal melanoma represents <5% of vulvar malignancies (discussed in the sixth article of this series), with Paget’s disease, sarcomas and basal cell lesions infrequently seen.

Stage I vulvar carcinoma is confined to the vulva. Stage II disease extends to adjacent perineal structures, such as the lower third of the vagina or urethra, or anus, with negative nodes. Stage III disease additionally involves inguinal nodes, with subtypes depending on the size and number of iliofemoral nodes involved and presence of extracapsular nodal extension. Stage IVA disease involves the upper third of the vagina, urethra, rectum or bladder, or with tumor that is fixed to the pubic bone or with ulcerated iliofemoral nodes. Of significance, and unlike other gynecological malignancies, Stage IVB disease involves intrapelvic lymph nodes as well as distant metastases.

The prognosis of Stage I epidermoid invasive vulvar carcinoma is good, at 85–90% five-year survival. Even in patients with inguinal node-positive disease, the five-year survival is 64% with lesions <2 cm and 43% with primary lesions >2 cm. The gold standard treatment of vulvar squamous cell carcinoma is surgery [61]. Advanced local disease is typically managed by a combination of surgery only, surgery plus radiation or combined chemoradiation, although the benefit of added radiation and chemotherapy in surgically resectable cases is controversial. The value of targeted therapies is under investigation.

The appropriate use of FDG PET-CT in the management of vulvar cancer has not been clearly established, and there are conflicting points of view in the literature. In early stage disease, the sensitivity and specificity for nodal involvement in the groin has been reported as 53% and 85% (Figure 20) [62]. This report suggests that, although positive lymph nodes on FDG PET are likely involved with tumor, a negative scan does not exclude disease. Alternatively, others have reported excellent performance of FDG PET-CT for groin nodes, with a sensitivity of 100% and specificity of 85%, with a maximum standardized uptake value (SUVmax) cutoff of 4.5, or a value of two times the mean SUV (SUVmean) of the liver [63]. However, this cut-off value has not been independently validated, prospectively. Sentinel lymph node localization studies remain the standard as an adjunct to surgical resection of the primary tumor. If there is a limitation to sentinel lymph node biopsy, FDG PET-CT may play a role in evaluating for pelvic nodal disease [64]. Prior to contemplated pelvic exenterative surgery, FDG PET-CT may play a role in excluding distant metastases [65]. The presence of Stage IVB includes the involvement of intrapelvic lymph nodes. In many circumstances, FDG PET-CT has been shown to be more sensitive than CT alone in identifying small positive nodes. Whether PET-CT upstages a significant number of patients with grade III disease or less, and the degree to which this affects outcome and management, is unclear but warrants further study.

### 2.7. Considerations of Pregnancy

Malignancies discovered during pregnancy create a risk not only to the mother but also the fetus if not managed carefully. If PET imaging is required for staging or other critical oncologic decision making, it should not be withheld because of concerns regarding radiation exposure. Nonetheless, care should be taken to keep radiation dose to the fetus as low as reasonably achievable in keeping with principles of ALARA. To date, no adverse effects on fetal outcome have been reported as a result of exposure due to a PET-CT scan performed during pregnancy. Nonetheless, strategies can be employed to reduce radiation dose to the fetus [66,67]. These include a low dose (mAs) CT or elimination of the CT and the use of a ^68^Ge transmission rod source for attenuation correction, which provides negligible exposure to the patient and fetus. Unless the patient has had a recent separately acquired CT that can be fused to the PET, the latter strategy is not practical in most cases. Additional methods to reduce the fetal dose are to use as low a dose of radiopharmaceutical as possible, often achievable in the range of 5 mCi for FDG, and to hydrate the patient and induce diuresis. The bladder should be emptied frequently after injection of the tracer, with consideration for the use of selective catheterization. This is because the major contributor of the dose to the fetus is the activity in the urinary bladder of the mother. Finally, dosimetric estimates should be calculated and reported as part of the medical record. The reader is referred to a document that provides excellent information on dosimetry, sources of exposure and strategies for employing the safest possible FDG PET scan in pregnant patients [68]. Utilizing these guidelines, PET scans can be performed without compromising fetal safety or diagnostic information. As a final note with regard to pregnancy, the role of FDG PET-CT for gestational trophoblastic neoplasia has been addressed in one small series and is said not to contribute additional information to that derived by conventional imaging modalities [69].

## 3. Urological Malignancies

### 3.1. Prostate Cancer

Prostate cancer is the most common cancer among men, excluding skin cancers. In the US, men have a 12.5% chance of being diagnosed with prostate cancer during their lifetime. Since it grows slowly in most men, patients often reach their life expectancy even with the disease. Despite this, prostate cancer is the second leading cause of cancer deaths among males [70]. In the early 1990s, widespread prostate specific antigen (PSA) screening among men resulted in a sharp rise in the incidence of prostate cancer, peaking in 1992. Since then, the US Preventative Services Task Force (USPSTF) has relaxed recommendations for screening, now leaving the decision to the patient and their primary care provider, with strong recommendations for PSA screening only for African American men and those with a family history of prostate cancer, which are factors associated with more aggressive disease [71]. As would be expected, this decrease in PSA screening has resulted in a decline in the overall incidence of prostate cancer, presumably from non-detection. However, in recent years, presumably because of a lack of early detection, there has been a progressive rise in the proportion of new prostate cancer cases that present with advanced disease [72]. This has presented new challenges and opportunities for PET-CT to contribute to the care of patients with prostate cancer.

Incidental identification of a hypermetabolic focus in the prostate on FDG PET is due to prostatitis or other benign causes in the vast majority of cases [73,74]. Heterogeneous metabolic activity can be seen with benign prostatic hypertrophy (BPH). FDG PET-CT has been shown to be ineffective in imaging moderately or well-differentiated prostate cancers, although more poorly differentiated tumors (Gleason score > 7) may be metabolically active (Figure 21). Prostate cancer with conversion to castrate resistance may also show increased metabolic activity on FDG PET-CT [74]. FDG PET may also prove useful in conjunction with PSMA PET in selecting patients most appropriate for future PMSA-targeted therapies, such as [^177^Lu]-PSMA-617 (Pluvicto^®^, AAA, Novartis, Basel, Switzerland), which was recently FDA approved [75,76]. Currently, FDG PET-CT does not typically play a role in prostate cancer, unless it is in cases where the tumor is highly dedifferentiated, such as initial small cell prostate cancer, or in heavily treated castrate resistant prostate cancer that undergoes dedifferentiation into prostate cancer with neuroendocrine differentiation or treatment emergent prostate cancer with small cell differentiation. In this case, PMSA PET-CT can be negative while FDG PET-CT is positive [75,76]. Nonetheless, incidental focal FDG uptake in the prostate, particularly in the peripheral zone, should prompt at least a serum PSA determination or further urological workup [73,74].

In recent years, novel prostate cancer PET-CT opportunities have emerged as critical imaging modalities in the management of prostate cancer. Currently, three new PET radiopharmaceuticals are FDA approved for prostate cancer and are widely available. These include Axumin^®^ ([^18^F]anti-1-amino-3-^18^F-fluorocyclobutane-1-carboxylic acid, also known as [^18^F]-fluciclovine or FACBC, Blue Earth Diagnostics, Oxford, UK) and two agents that target prostate specific membrane antigen, PYLARIFY^®^ ([^18^F]-piflufolastat, also called DCFPyL or Pyl, Lanheus/Progenics, Millerica, MA, USA) and LOCAMETZ^®^ ([^68^Ga]-PSMA-11, PSMA-HBED-CC, gozetotide, Advanced Accelerator Applications, Millburn, NJ, USA) [77,78,79,80,81]. Additionally, ^11^C choline is FDA-approved and has been extensively used at Mayo Clinic (Rochester, MN, USA), but requires an on-site cyclotron and is limited by a very short half-life (20.38 min) [82]. Additionally, [^18^F]-NaF PET-CT has been used as a PET-CT alternative to conventional bisphosphonate scintigraphy but has not been widely adopted to replace the conventional scintigraphic bone scan [83]. For purposes of this review, [^18^F]-DCFPYL will be referred to as [^18^F]-PSMA and [^68^Ga]-PMSA-11 will be referred to as [^68^Ga]-PSMA. PSMA PET-CT will refer to either labeled construct, interchangeably.

[^18^F]-fluciclovine was the first new generation prostate imaging PET radiopharmaceutical to be FDA-approved and widely available. It has now been in use for several years for its sole approved indication: identification of sites of disease in men with biochemical recurrence following definitive treatment. Biochemical recurrence is categorized as PSA of >0.2 ng/mL following prostatectomy or a PSA > 2.0 ng/mL above nadir following external beam or brachytherapy radiation treatment. In addition to biochemical recurrence following definitive therapy, the two PSMA targeting tracers, [^18^F]-PSMA and [^68^Ga]-PSMA, are also FDA approved for up-front assessment of metastatic disease in men eligible for definitive treatment. This typically will apply to those with unfavorable intermediate-risk, high risk and very-high-risk prostate cancer without confirmed metastatic disease. It could also include patients who have been under active surveillance for lower-risk disease who are now suspected as having developed higher risk disease based on a rapidly increasing PSA or other clinical factors. [^68^Ga]-PSMA is additionally approved for selection of patients eligible for ^177^Lu PSMA-617 (Pluvicto^®^). The risk categories for prostate cancer are defined by NCCN guidelines and outlined in Table 1 [84]. Of note, Gleason scoring has recently been reclassified as Grade Groups by NCCN [84] (Table 2). For up-front staging, the evidence and increasing opinions support the use of PSMA (either as F-18 or Ga-68 labeled) PET-CT as the first line imaging study over CT [85,86]. As stated by the NCCN consensus panel for prostate cancer, “Because of the increased sensitivity and specificity of PSMA-PET tracers for detecting micrometastatic disease compared to conventional imaging (CT, MRI) at both initial staging and biochemical recurrence, the Panel does not feel that conventional imaging is a necessary prerequisite to PSMA-PET and that PSMA- PET-CT or PSMA-PET/MRI can serve as an equally effective, if not more effective front-line imaging tool for these patients” [84].

[^18^F]-fluciclovine is a radiolabeled synthetic L-leucine analog that is transported across cell membranes by LAT-1, ACST2 and SNAT2 amino acid transporters that are upregulated in prostate cancer cells. Early localization at sites of prostate cancer occurs, but the activity washes out relatively quickly, requiring initial PET imaging of the pelvis and then scanning upward, to begin within 4 min after injection. Determination of whether lesions are positive for prostate cancer, equivocal or likely benign on fluciclovine PET-CT typically requires a comparison of activity in a lesion to that in reference tissues, specifically, blood pool and normal vertebral bone marrow. These reference guidelines are provided for the intact prostate (Table 3), seminal vesicles (Table 4), the post-surgical prostatectomy bed (fossa) (Table 5), and lymph nodes (Table 6). Due to normal heterogeneity in marrow activity and interspersed fatty and red marrow in older patients, our recommendation is to use SUVmax, rather than SUVmean, for the bone marrow reference. [^18^F] fluciclovine is limited in detection of bone metastases from prostate cancer. Some lesions are strongly positive, others show no increased uptake over background lesions. 

[^18^F]-PSMA and [^68^Ga]-PSMA are now both widely available for imaging prostate cancer. These agents bind to the external domain of the prostate specific membrane antigen type II transmembrane protein. Expression is highest in prostate cancer but it can also be expressed by other solid tumors. Uptake of the PSMA agents in BPH is typically relatively low, as opposed to the high uptake in BPH seen with [^18^F] fluciclovine. [^177^Lu]-PSMA-617 (Pluvicto^®^) was recently FDA approved for treatment of metastatic castrate resistant prostate cancer along with its corresponding equivalent imaging agent, [^68^Ga]-PSMA. Confirmation of PSMA expression by [^68^Ga]-PSMA PET-CT was important in the initial [^177^Lu]-PSMA-617 trials because 16% of subjects were excluded because of poor uptake on PSMA PET-CT [88]. As discussed above, some cases of advanced castrate resistant metastatic prostate cancer may show decreased PSMA expression. Therefore, [^68^Ga]-PSMA PET-CT will be important in selecting patients for [^177^Lu]-PSMA-617 treatment. Of note, [^18^F]-PSMA is not approved for selection of patients for [^177^Lu]-PSMA-617.

Two interpretative schemes that serve as models for structured reports have been proposed for diagnostic PSMA PET-CT, including a system proposed by a single institution (Johns Hopkins University) (Table 7) [89] and a system used in the PROMISE trial (Table 8) [90]. For [^68^Ga]-PSMA identification of bone metastases, it has been additionally proposed that positive lesions be defined as having a lesion:normal tissue SUVmax ratio of 2.2 for blood pool and 4.1 for normal bone [91]. However, neither these indices nor the structured reporting classifications have been prospectively and independently validated in large series and, at this point, are best regarded as consensus recommendations. The criteria listed below can be used for both [^68^Ga]-PSMA and [^18^F]-PSMA PET-CT.

Head to head comparisons between [^18^F]-fluciclovine and PMSA PET-CT for identification of prostate cancer have been reported, although most involve comparison of [^18^F]-fluciclovine and [^68^Ga]-PSMA [92]. Since urine activity is very high with PSMA PET, and low to moderate with [^18^F]-fluciclovine, [^18^F]-fluciclovine PET-CT has an advantage over PSMA for detection of tumor in the post-prostatectomy surgical fossa (Figure 22). For this reason, we recommend that patients undergo either PSMA or [^18^F]-fluciclovine PET-CT be scanned with a CT protocol that includes intravenous contrast. This enables identification of some small enhancing areas of recurrence in the fossa that might be otherwise missed because of surrounding urine activity. Good hydration for PSMA PET scans and voiding no less than one hour prior to imaging (to allow for some dilution of urine in the bladder) is recommended. For detection of pelvic nodes and extra-pelvic disease, PSMA PET has been shown to be superior to ^18^F fluciclovine. For example, in one report of 50 patients with biochemical recurrence and PSA levels of 0.2 to 2.0 ng/dL, PSMA PET has significantly greater detection rates over [^18^F]-fluciclovine PET-CT for overall disease (56% vs. 26%), as well as pelvic nodal (30% vs. 8%) and extra-pelvic (16% vs. 0%) sites of prostate cancer [92]. For evaluation of patients whose prostate is intact, fluciclovine PET-CT presents a disadvantage, because uptake of the tracer in benign prostatic hypertrophy is considerable and may mask the presence of prostate cancer (Figure 23). In this regard, PMSA PET-CT is at an advantage, because uptake by normal prostate, or BPH, is minimal, allowing easy identification of sites of prostate cancer in the prostate itself. In the identification of the tumor in the intact prostate, PSMA also offers an advantage over MRI when BPH is present (Figure 24).

PSMA PET-CT has been shown to be superior to [^18^F]-fluciclovine PET-CT in the detection of bone metastases [93]. [^18^F]-fluciclovine PET-CT is complicated by the fact that many osseous prostate cancer metastases are not fluciclovine-avid (Figure 25 and Figure 26). Both [^18^F]-fluciclovine and PSMA PET-CT can show uptake at sites of fibrous dysplasia and vertebral hemangiomas [94,95]. Fluciclovine PET-CT has been reported to be positive in osteoid osteoma [96]. For any atypical osseous lesion identified by either [^18^F]-fluciclovine or PSMA PET-CT, additional confirmatory imaging, such as with MRI or close longitudinal observation, should be recommended.

Few head-to-head comparisons between [^18^F]-PSMA and [^68^Ga]-PSMA have been performed. However, in one small series, [^18^F]-PSMA outperformed [^68^Ga]-PSMA in intensity of uptake within sites of tumor, in higher tumor:background ratios, and in number of very small lesions detected [97]. This difference is supported by consideration of the physical characteristics of the two isotopes. The lower positron energy (keV) and shorter positron range (continuous slowing down approximation in water, CDSA) of F-18 (250 keV, 0.062 cm CDSA), compared to that for Ga-68 (830 keV, 0.337 cm CDSA), should equate to nearly a two-fold greater PET resolution for F-18 (1.4 mm) than for Ga-68 (2.4 mm) in all directions at center [98]. Furthermore, the administered activity of [^68^Ga]-PSMA is approximately 5 mCi, and that for [^18^F]-PSMA is 10 mCi. After an uptake interval of approximately 60 min, 2.5 times more activity is present with [^18^F]- compared to [^68^Ga]-PSMA due to the shorter half-life of Ga-68. With greater positron flux, resolution and activity, this will translate to less noise and greater lesion detectability with [^18^F]-PSMA than with [^68^Ga]-PSMA, assuming all other factors are equal [99]. With prostate cancer, this difference may be critical in that the smallest possible lesions must be identified early so that appropriate treatment can be initiated.

Both PSMA PET agents are approved for assessment of metastatic disease in patients eligible for both primary definitive or salvage treatment. As such, opportunities exist to address the primary tumor in the prostate itself prior to treatment. Typically, gross tumor volume and evidence of extracapsular extension is assessed by multiparametric MRI (mpMRI). However, there is evidence that PSMA PET-CT may compare favorably to mpMRI in this regard [100]. This is particularly true when brachytherapy seeds or metal artifacts created by total hip prostheses are present, which create significant artifact on MRI (Figure 27). [^68^Ga]-PSMA PET-CT has been shown to be superior to mpMRI in assessing extracapsular extension and equivalent to mpMRI in assessment of seminal vesicle involvement [101]. In comparison to histopathology on a slice by slice basis, it has been reported that mpMRI underestimates tumor volume significantly when compared to PSMA PET-CT [102]. Due to the potential superiority of PSMA PET-CT over MRI in imaging disease in the intact prostate, it is reasonable to imagine that an additional application of PSMA PET as a substitute for MRI in patients being initially evaluated for possible prostate cancer or a reasonable alternative when MRI is contraindicated or likely to have extensive artifacts, such as with metallic hip prostheses. However, PMSA PET-CT is not FDA-approved for the diagnosis of prostate cancer prior to biopsy confirmation.

Multi-parametric MRI, followed by biopsy and then a conventional bisphosphonate bone scan, is most typically performed for patients with newly diagnosed prostate cancer. The addition of PMSA PET-CT after mpMRI and bone scan has been shown to change treatment in 22% of men [103]. It is tempting to postulate that a PMSA PET-CT scan could substitute for a conventional bone scan. However, even with an initial PSMA PET-CT scan, longitudinal surveillance of patients will likely include bone scintigraphy, and a baseline study is indicated as a point of reference. Furthermore, PSMA PET-CT has not been FDA-approved for longitudinal surveillance in men with known metastatic disease. For these reasons, it is likely that multimodality mpMRI, PSMA PET-CT and a bone scan will all likely be required in men with initial diagnosis of Grade group 3-5 prostate cancer. The availability of multi-modality imaging in prostate cancer can also be leveraged in the development of radiomic feature classification of prostate cancer. It has been suggested that prostate cancer may serve as an ideal platform for the development of artificial intelligence (AI) and machine learning techniques using features derived from both PET-CT and MRI to diagnose and determine the grade of prostate cancer [104,105]. However, these applications have yet to be employed in routine practice.

There are rare instances when PMSA PET-CT may show lower than expected uptake in the primary tumor in the prostate compared to distant metastases (Figure 28). The reason for this is unclear but may be related to cellular internalization of the PSMA protein. It has been reported that PSMA PET-CT is more strongly positive in higher Gleason Grade or Grade Group tumors and may be more accurate than mpMRI in defining large volume disease [106]. PMSA PET activity is generally low in BPH, which offers a substantial advantage over [^18^F]-fluciclovine, for which there is a significant overlap between activity due to BPH and that due to prostate cancer (Figure 23 and Figure 24). In identifying recurrence within the intact prostate in patients who have had external beam or brachytherapy, [^18^F]-fluciclovine PET-CT is highly sensitive (approaching 100%) but very poorly specific, reported to be as low as 11% [107]. (Figure 29).

It has been reported that the lower the PSA level, the less likely it will be to have a positive fluciclovine scan, such that with PSA levels < 0.8 ng/dL, positivity for any lesion is only approximately 41% [108]. It has also been shown that the optimal PSA cut-off for performance of [^68^Ga]-PMSA PET-CT is 1.24 ng/mL for post prostatectomy patients, and 5.75 ng/mL post radiation therapy [109]. However, even with low PSMA (<0.2 ng/dL), there are, nonetheless, a substantial number of patients (63.6%) with positive scans [110]. The lower PSA limit for PSMA PET-CT that results in a positive scan has not been established, but there are clear advantages to finding sites of disease at the earliest possible point. If a PSMA PET-CT scan is negative, it can always serve as a baseline for future evaluation.

The question of whether androgen deprivation therapy (ADT) decreases diagnostic accuracy of [^18^F]-fluciclovine or PSMA PET is an important issue. Information regarding the effect of concurrent ADT on performance of [^18^F]-fluciclovine PET-CT is lacking. However, there is significant evidence that long-term ADT may suppress uptake of PSMA ligand by sites of tumor involvement [111,112,113,114]. Conversely, short term ADT may increase uptake. Short term (8–11 days) low dose treatment with the gonadotropin releasing hormone receptor antagonist Degarelix (Firmagon^®^) enhances uptake on PMSA PET in areas of prostate cancer involvement of lymph nodes and bone metastases [115]. Similarly, short term (7 day) treatment with the androgen-receptor blocker Apalutamide (ARN-509, ERLEADA^®^) has also been shown to increase uptake of PMSA ligand in an animal model as well as in a human with metastatic prostate cancer [116]. There is no information as to the interval of long-term ADT withdrawal necessary prior to performance of a PSMA PET-CT. In our anecdotal experience, PSMA PET-CT scans are often positive in patients with progressively rising PSA levels, despite long term ADT, supporting resistance to ADT (castrate resistance).

There are several critical principals in imaging for prostate cancer that apply both to [^18^F]-fluciclovine and PSMA PET-CT. For patients with biochemically recurrence post-prostatectomy, accurate identification of the site of recurrent disease is critical in that it influences decisions and fields for salvage radiation. If disease is distantly metastatic, ADT or other systemic therapies are more conventional, although some trials may target specific sites of oligometastatic disease by surgery or radiation. [^18^F]-Fluciclovine PET-CT has been shown to affect a change in salvage radiation therapy management of a significant proportion of patients with biochemical recurrence post prostatectomy [117]. Of note, urine activity can complicate detection of small foci of recurrence in the prostate fossa, which is more of a problem with PSMA than [^18^F]-fluciclovine PET-CT.

Extra-prostatic extension into extra-nodal structures of the pelvis is a serious complication of prostate cancer and attention to the possibility of this should be given when reading a PET-CT scan. Extra-prostatic spread can be subtle or grossly evident (Figure 30 and Figure 31). In addition, when interpreting a PET-CT scan for prostate cancer, it is important to note when there is capsular abutment of the tumor of >1cm. This creates a risk of extracapsular extension even though it may not be visually appreciable on PET-CT. Seminal vesicles represent an early route of extra-prostatic spread of the primary tumor and may be involved in recurrent disease in patients with an intact prostate. From there, spread can occur to the neurovascular bundle or along the vas deferens (Figure 32). It is important to know that seminal vesicles are typically removed with radical prostatectomy. Vas deferens are typically not resected, except at their connection to the seminal vesicles. Correct identification of the neurovascular bundle, or pedicle, which lies near the bed of the seminal vesicles is critical (Figure 33). Soft tissue fullness and focal uptake in this region has significance in that it represents a route of systemic spread of disease as well as a structure critical for maintenance of erectile function and continence. From the neurovascular pedicle, perineural spread of prostate cancer may extend along branches of the inferior hypogastric plexus that are closely associated with the mesorectal and Denonvilliers fascia. Recognition of this pattern of spread as a potential radiation therapy target is critical for local and symptomatic control (Figure 34).

Nodal spread of disease is considered regional if it involves branches of the internal or external iliac vessels below the bifurcation, as well as presacral, periprostatic and perirectal nodes. Common iliac and periaortic/pericaval nodes, to the level of the renal vessels, represent the next order of spread, and are considered non-regional nodes. Involvement of upper abdominal retrocrural as well as left medial supraclavicular nodes typically herald distant metastatic spread. It is critical to recognize lumbar, celiac, sacral and stellate ganglia, which are typically mildly PET-positive both on [^18^F]-fluciclovine and [^68^Ga]- or [^18^F]-PSMA PET-CT and can be easily confused with early involvement of lymph nodes. These ganglia are typically bilateral structures but may not be perfectly symmetrical (Figure 35).

Both of the FDA-approved and commercially available PSMA ligands may show increased uptake of radiotracer in a wide range of other solid tumors, both benign and malignant, including high grade glioma, breast cancer, meningioma, hepatocellular carcinoma, renal cell carcinoma, lung cancer, GIST tumor, urothelial carcinoma, hepatic hemangiomas, testicular/embryonal and others [118]. [^18^F]-Fluciclovine PET-CT has also been shown to have increased uptake in cutaneous and head and neck squamous cell carcinoma, meningiomas and melanoma, both lobular and ductal breast cancer, lymphoma, multiple myeloma, colon cancer, carcinoid tumors and lung cancer, nerve sheath tumors and thymoma [119]. Most of the time, the magnitude of uptake on PSMA or [^18^F]-fluciclovine PET-CT in non-prostatic sites of tumor is less than would be expected for prostate cancer. However, for example, with PSMA ligand uptake in GIST tumors (Figure 36), and meningiomas, uptake can be intense. It is fair to say that with either fluciclovine or PMSA PET-CT, any atypical or unexpected site of uptake, or uptake in a lesion that is larger than would be expected for the degree of PSA elevation, should be regarded in light of the possibility that it could represent either benign or malignant tumors of non-prostatic origin and should prompt further investigation by imaging, tissue sampling or longitudinal observation (Figure 36 and Figure 37). As experience is gained using PSMA PET-CT, which is likely to ultimately replace [^18^F]-fluciclovine PET-CT, clearer guidelines will be developed for image interpretation as well as patient selection and preparation. Expanded applications of PSMA PET-CT to include other PSMA-rich tumors will likely also be developed.

### 3.2. Urothelial Carcinoma

Urothelial carcinoma can involve the upper tract (renal calyces, pelvis and ureter) or the lower tract (urinary bladder). There are a number of histological variants of urothelial carcinoma. A minority of tumors include squamous cell carcinoma and adenocarcinoma. Bladder carcinoma accounts for 90–95% of urothelial carcinomas and is the third and fourth most common malignancy in women and men, respectively [120]. Urothelial carcinoma typically presents with painless hematuria and occasionally dysuria or urgency. Flank or pelvic pain typically occurs as the disease becomes more advanced. The diagnosis of urothelial carcinoma is made by urine cytology, cystoscopy or, with upper tract disease, CT urography.

Non-muscle invasive bladder cancer is typically managed by transurethral resection (TURBT), which is often followed within hours by intravesicular chemotherapy for low risk disease or intravesicular immunotherapy with Bacillus Calmette–Guerin (BCG) for higher risk disease [121]. Muscle invasive bladder cancer is typically managed by radical cystoprostatectomy for men, or cystectomy and anterior pelvic exenteration in women, with bilateral pelvic or extended lymph node dissection and creation of a urinary diversion. Neoadjuvant or adjuvant chemotherapy may be used in some patients. Patients with metastatic bladder cancer with FGFR-2 or -3 mutations who fail platinum-containing chemotherapy may benefit from Erdafitinib.

Patients with upper tract urothelial carcinoma account for approximately 10% of urothelial carcinomas. Patients with concomitant upper and lower tract disease represent a similar percentage of those with urothelial carcinoma and are more common with high grade tumors. Those with primary ureteral involvement have a worse prognosis than those with renal pelvic disease. Standard treatment for both ureteral and renal pelvis urothelial carcinoma is radical nephroureterectomy. Nephron sparing surgery for low grade ureteral tumors may be employed in patients with critical renal insufficiency or a single kidney. Percutaneous or retrograde administration of BCG is commonly employed. Muscle invasive or locally advanced disease at the time of surgery is typically treated with neoadjuvant or adjuvant gemcitabine and cisplatin chemotherapy. Metastatic or node positive disease is also treated with gemcitabine and either cisplatin or (in cases of renal insufficiency) carboplatin. Check-point inhibitors may be of benefit with chemotherapy-resistant metastatic disease.

Metabolic activity on FDG PET-CT in urothelial carcinoma is typically high (Figure 38). Due to high urine activity, FDG PET-CT is challenging in evaluation of the primary urothelial tumor either within the bladder or upper tract (Figure 39). However, FDG PET-CT has been shown to change management in urothelial carcinoma in 40% of patients with upper tract disease and 74% of those with urothelial bladder carcinoma [122]. In muscle invasive bladder cancer, the sensitivity of FDG PET-CT has been shown to be superior to that of CT alone for detection of distant metastases (54–87%, 41%), although the specificity of both is high and similar (90–97%, 98%) [123]. For identification of nodal disease, the sensitivity of FDG PET-CT is similarly superior to that of CT alone (52%, 38%), and the specificity is also high and similar for both (91%, 92%). Criteria for assessing therapeutic response have not been established for FDG PET-CT of urothelial carcinoma. For recurrent disease, the sensitivity of FDG PET-CT is 87–92%, and it results in a change in management in 40% [123]. The NCCN broadly supports the use of FDG PET-CT in muscle invasive bladder cancer for staging when metastatic disease is not known, when metastatic disease (soft tissue or bone) is suspected and for follow-up with or without cystectomy (Figure 40) [124]. It should be noted that metastatic spread of urothelial carcinoma in the pelvis and retroperitoneum is often a diffuse, infiltrative-appearing process, rather than focal hypermetabolic nodular or discrete nodal regions (Figure 40).

Published reports regarding the impact and appropriate role of FDG PET-CT with upper tract urothelial carcinoma are limited. NCCN does not separately address upper tract urothelial carcinoma but cautions that FDG PET-CT should not be used to assess whether upper tract disease is also present [124]. The value of FDG PET-CT is best recognized for assessment of metastatic disease, where PET-CT outperforms CT in sensitivity on a lesion-based analysis (85% vs. 50%) but is comparable on a patient-based analysis [125]. In identification of the presence of upper tract urothelial carcinoma, however, PET-CT has been reported to identify abnormalities in 83% of patients with proven upper tract disease, with a positive predictive value (PPV) of 95% [126]. There is little information regarding the value of FDG PET-CT in assessing response to treatment with urothelial carcinoma, but it may be of use in selected cases with a persistent mass following treatment (Figure 41). Therefore, PET-CT may prove to be of value when urine cytology is non-diagnostic, and attention to the collecting system on FDG PET-CT scans that are conducted for other indications may occasionally result in identification of an unsuspected urothelial carcinoma.

### 3.3. Primary Testicular Cancer

Testicular cancer can occur as a primary tumor, metastatic disease or lymphoma. Primary testicular neoplasms broadly consist of two categories, germ-cell and non-germ cell tumors. Lymphoma and metastases can occur in the testis, accounting for 10% of testicular tumors. However, lymphoma is the most common testicular cancer in patients greater than 60 years of age and is bilateral in 50% of patients. Metastases to the testis most commonly occur from prostate, lung, renal cell, GI cancers and melanoma [127].

Non-germ cell (gonadal stromal cell) tumors are benign in 90% of cases. These consist of Leydig, Sertoli and Granulosa cell tumors, gonadoblastoma and mixed cell tumors. Gonadoblastoma occurs in abnormal gonadal tissue in patients with disorders of sex development. Since there is some risk of malignant transformation, these tumors are typically surgically removed. FDG PET-CT could theoretically play a role in evaluation of metastatic lesions in this entity, but the paucity of literature does not allow an assessment of the potential value of PET in patients with non-germ cell testicular tumors.

Primary testicular germ cell tumors are the most common solid malignancy in men age 20–35. There are two major categories of germ cell tumors: pure seminoma and non-seminomatous. Non-seminomatous tumors consist of teratoma/teratocarcinoma, embryonal/choriocarcinoma and mixed tumor. These typically display elevated serum tumor markers, including alpha fetoprotein (aAFP) and human chorionic gonadotropin (hCG). Seminomas have the best prognosis and typically have no elevated serum markers. Mixed seminoma and non-seminoma tumors, or seminomas with elevated serum cancer markers, are treated as non-seminomatous because of a worse prognosis. The overall prognosis with germ cell tumors is good, even with metastases, with an overall five-year survival rate of 95% (72.8% with metastatic disease) [128]. The spread of the tumor is both by lymphatic and hematogenous spread, with regional nodal spread to the pelvis and peritoneum, and with it distantly metastasizing primarily to the lungs and mediastinal nodes.

Patients with germ cell tumors typically present with a painless testicular mass or nodule. Sonographic characterization, serum markers, CT staging and radical orchiectomy are the standard management for germ cell tumors. Stage I seminomas can often be treated by orchiectomy only, followed by surveillance, but options for higher risk tumors typically include post-adjuvant radiation or chemotherapy. Surveillance, consisting of serum markers and CT, occurs at prescribed intervals for up to 10 years, since late recurrences do occur. Recurrence, depending on risk features, is typically treated with salvage surgical resection of a solitary site of recurrence, chemotherapy (conventional or high dose) and autologous stem cell transplant. With residual disease after chemotherapy, nerve-sparing retroperitoneal lymph node dissection (RPLND) is recommended. If disease is found, chemotherapy is administered.

The use of FDG PET-CT is not recommended for initial staging or restaging of germ cell tumors. For seminomatous germ cell tumors, FDG PET-CT can be used to assess treatment response but has several significant limitations. There is significant variability in the magnitude of uptake in sites of seminoma tumors (Figure 42). Typically, larger tumors of those widely metastatic are more metabolically active. For residual masses < 3 cm, PET-CT is not indicated because of the low rate of viable cells. For lesions > 3cm in pure seminomas, FDG PET-CT is recommended post treatment [129,130]. It has a negative predictive value of 94–96%. However, the positive predictive value is poor, at 23–50%. There is often demonstrable metabolic activity post treatment, even when a viable tumor is no longer present (Figure 43). FDG PET-CT, if performed, should be delayed for at least six weeks after completion of treatment, if possible. Even with a delay in imaging following treatment, if a post treatment scan is positive in the face of a reduction in size of the lesion, a follow-up scan might be contemplated rather than moving directly to RPLND because of a high degree of complications of anejaculation and lymphatic leak. In short, in the management of germ cell tumors, a negative scan following treatment has a high negative predictive value, and the modality may be of value in specific cases for problem solving. However, an increase in metabolic activity and size of a tumor are indicative of progression (Figure 44). In patients with suspected recurrent germ cell tumor, FDG PET-CT has been shown to provide prognostic value in predicting progression free survival (PFS) and overall survival (OS). In addition, FDG PET-CT impacted management, primarily in terms of restaging, in 23% of patients.

For non-seminomatous germ cell tumors, PET-CT is not indicated because there is a clear survival benefit with empirical RPLND in patients without distant metastatic disease, even in low stage tumor [131]. For teratomas, it is commonly agreed that even PET-negative lymph nodes >1 cm should be surgically removed. Nonetheless, for non-seminomatous germ cell tumors, FDG PET-CT may be useful in selected cases in evaluating for metastatic disease (Figure 45).

### 3.4. Renal Cell Carcinoma

Renal cell carcinoma (RCC) accounts for 3% of adult malignancies and 90–95% of all kidney neoplasms [132,133]. Risk factors include smoking, obesity, hypertension and chronic renal failure. Histologic types of RCC include clear cell (75–85%), papillary (10–15%), chromophobe (5–10%) and rare collecting duct variants. Oncocytomas, comprising 3–7% of renal neoplasms, are typically benign but occasionally can be pre-malignant or malignant. RCC is typically clinically silent until advanced. RCC spreads by local and vascular invasion and by hematogenous spread, primarily to the lungs and bones. Localized disease can be cured by surgical resection, which may also improve survival, even in patients with metastatic disease [134]. Small tumors may also be managed by thermo- or other ablative techniques. Surgical resection or ablation of sites of metastatic disease, even if multiple, also improves survival. Advanced disease is typically treated by combinations of anti-angiogenic, immunotherapy, molecular-targeting agents and cytokines [135].

By FDG PET-CT, metabolic activity in RCC is typically similar to, or only slightly greater than, that in the normal renal parenchyma, and much less in activity than the urine, making detection by FDG PET-CT a challenge unless the lesion is exophytic, a property that is also true for oncocytoma (Figure 46 and Figure 47). The distinction between oncocytoma and renal cell cancer has been a challenge for all imaging modalities. A novel PET radiopharmaceutical under development, [^89^Zr]Zr-DFO-girentuximab, shows uptake in renal clear cell carcinoma but not in oncocytomas. Adding [^89^Zr]Zr-DFO-girentuximab to the conventional imaging approach increases lesion detection from 56% to 91% [136]. [^99m^Tc]-sestamibi SPECT-CT also shows markedly increased uptake in oncocytoma and hybrid oncocytoma/chromophobe renal cell carcinomas but not in renal clear cell carcinoma [137]. Ultimately, a hybrid approach using [^89^Zr]Zr-DFO-girentuximab and [^99m^Tc]-sestamibi SPECT/CT may prove to be optimal in characterizing renal masses [138]. It should also be noted that renal cell carcinomas have been reported to show uptake on PSMA PET. However, this has not been extensively studied at this point [139]. [^18^F]-fluciclovine has been shown to have increased uptake in renal papillary cell carcinomas but not in renal clear cell carcinoma [140].

Conventional imaging for renal cell carcinoma includes contrast-enhanced CT and MRI. Lung and bone metastases may be somewhat more metabolically active than the primary tumor, but are greatly variable and are typically evident by CT. As such, FDG PET-CT offers no specific advantage in the imaging of RCC over CT or MRI. As stated by the NCCN, “The value of PET in RCC remains to be determined. Currently, PET alone is not a tool that is standardly used to diagnose RCC or follow for evidence of relapse after nephrectomy”, an opinion that is further supported by other reports in the literature [135,141]. Nonetheless, there is variability in the uptake of FDG by renal cell carcinomas, and FDG PET-CT has been suggested to potentially a role in the evaluation of metastatic disease and in predicting survival and progression [142,143]. Sarcomatoid transformation can rarely occur in all histologic types of RCC and is very aggressive, with a poor prognosis. This condition is typically intensely FDA-avid, and a widely metastatic or markedly invasive tumor can suggest the diagnosis (Figure 48) [144]. However, intensely FDA-avid renal masses are more likely to represent lymphoma or metastatic disease (Figure 49 and Figure 50).

The World Health Organization (WHO)/International Society of Urologic Pathologists (ISUP) grading system for renal cell carcinomas of papillary and clear cell types has been shown to have significant prognostic significance. The use of radiomic features to characterize renal masses, predict prognosis and grade tumors has been a focus of research. CT-based radiomics have been identified as being predictive of WHO/ISUP grade [145]. FDG PET-CT has also been studied as a predictor of WHO/ISUP grade as well. A tumor:liver metabolic ratio of >1.63 and the presence of pre-operative tumor thrombus on FDG PET-CT have been shown to identify high grade tumors [146]. Assessment of vascular invasion is an important feature in staging, surgical planning and prognosis. However, vascular invasion, unless bulky, may be difficult to appreciate on FDG PET-CT without intravenous iodinated contrast for the CT component of the PET-CT (Figure 51). Future development of novel targeted therapies for renal cell carcinoma will open the door to pre-treatment characterization of tumor phenotypes to facilitate personalized treatment [147].

### 3.5. Penile Cancer

Penile cancer, most of which consists of squamous cell carcinomas, is rare in western cultures but proves fatal in 21% of patients [131]. Risk factors for penile carcinoma include HPV, high-risk or early sexual activity, lack of circumcision and chronic lymphocytic leukemia (CLL). Diagnosis is often delayed for many reasons until the disease is advanced. Treatment of limited lesions of the glans, urethreal meatus or prepuce (foreskin) may be accomplished by Moh’s surgery, circumcision (if confined to the prepuce), radiotherapy or by topical chemotherapy. Invasive disease may be managed by resection with or without penile reconstruction. Chemotherapy is usually employed when inguinal or more distant metastases are involved.

The appropriate role of FDG PET-CT in penile cancer is somewhat uncertain. The primary tumors are easy to miss and urine contamination may further limit detection by FDG PET-CT. Both the sensitivity and specificity of FDG PET-CT in identifying the primary lesion have been reported as 75% [148]. First order drainage is typically limited to inguinal nodes. The value of FDG PET-CT in identifying inguinal nodal involvement is limited with nodes that are not enlarged to palpation (cN0), with a sensitivity of approximately 56% [149,150]. Clinically enlarged inguinal nodes (cN+) are typically identifiable by PET-CT, with a sensitivity of around 96%. There are insufficient data to allow an assessment of the value of FDG PET-CT in identifying distant metastases or in evaluating response to treatment. However, most squamous cell carcinoma metastases, including those of the penis, are very FDG-avid (Figure 52 and Figure 53). Therefore, the NCCN takes a permissive role with respect to the use of FDG PET-CT in penile cancer. With non-palpable nodes, the NCCN endorses the use of either CT, MRI or PET-CT for evaluation of intermediate to high risk (T1b or T2) lesions [151]. NCCN also recommends that FDG PET-CT be considered for risk stratification with palpable inguinal adenopathy, for assessment of metastatic disease and for evaluating treatment response. However, the NCCN makes no recommendations regarding the use of FDG PET-CT for surveillance. NCCN recommends that CT imaging be performed with contrast. Metastatic disease to the penis from other malignancies is rare but does occur, typically from prostate, bladder, lung, stomach, colon or skin (Figure 54). However, aggressive tumors of other types may metastasize to the penis as well (Figure 55) [152].

## 4. Conclusions

The role of PET-CT, and the diversity of radiopharmaceuticals available for tumor imaging, is an ever-evolving field. The degree to which PET-CT contributes to the overall clinical management of patients with cancer is dependent on many important factors that require dedication, ongoing self-education and good communication between the providers and imaging professionals. An understanding of the normal biodistribution and physiologic parameters that may alter the distribution of the specific radiopharmaceutical being used is needed in order to accurately interpret the scans. This is particularly true with newly-approved PET radiotracers, with which most imaging groups have little experience. Pitfalls, artifacts and normal variants that may simulate disease may vary as a function of the radiopharmaceutical used and the organ system examined. The maintenance of a knowledge of the literature is critical to for ongoing professional improvement and education in accurate image interpretation. This typically involves both a familiarity with the radiology and nuclear medicine literature along with critical reports and the outcome of clinical trials in oncology-specific publications. A consistent manner of reporting, preferably using nationally consensus-adopted structured reporting strategies, is highly recommended. An in-depth knowledge of the specific type of cancer being evaluated, an awareness of the typical spectrum of clinical and imaging findings, the expected patterns of spread of disease, the staging systems specific to the type of tumor being assessed, the therapeutic options and the typical complications of those therapies should be understood. Finally, and perhaps most importantly, it is critical to have knowledge of the patient’s history and the specific questions being asked by the referring providers. If diligence is applied to ensure that all of these requirements are met and maintained, the radiologist or nuclear medicine practitioner becomes a true member of the interdisciplinary health care team. In addition, the referring providers should understand the limitations and advantages of each type of PET-CT scan for the specific type of tumor for which the referral for imaging is made. The providers should communicate well with the imaging professionals and ask specific questions that will guide the imagers in providing the most relevant and accurate information.

## Figures and Tables

**Figure 1 cancers-14-03000-f001:**
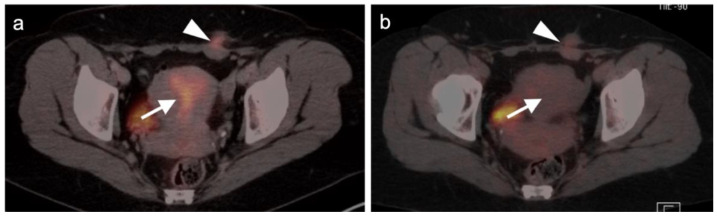
Physiologic hypermetabolism in the endometrium and in a scar endometrioma in a woman who underwent a prior Caesarian-section. FDG-PET-CT images of the pelvis. (**a**) Normal endometrial hypermetabolism (white arrow) occurs during the first 4 days of the menstrual cycle and during mid-cycle. An endometrial implant on the anterior abdominal wall also shows similar physiologic uptake; (**b**) A second FDG PET-CT scan performed off-cycle shows resolution of the metabolic activity both in the endometrium and in the scar endometrioma (subsequently biopsy proven).

**Figure 2 cancers-14-03000-f002:**
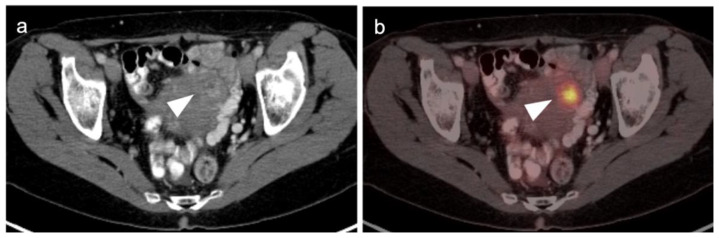
Benign leiomyoma (uterine fibroid). (**a**) Axial contrast-enhanced CT of the pelvis shows vague enhancement of a uterine nodule (white arrowhead); (**b**) Axial FDG PET-CT images of the pelvis at the same level shows a hypermetabolic uterine nodule, which was felt by additional imaging and stability over time to be a benign uterine leiomyoma. Metabolic activity within uterine leiomyomas can be extremely variable. Some are hypermetabolic for unknown reasons, as is shown here (white arrowhead).

**Figure 3 cancers-14-03000-f003:**
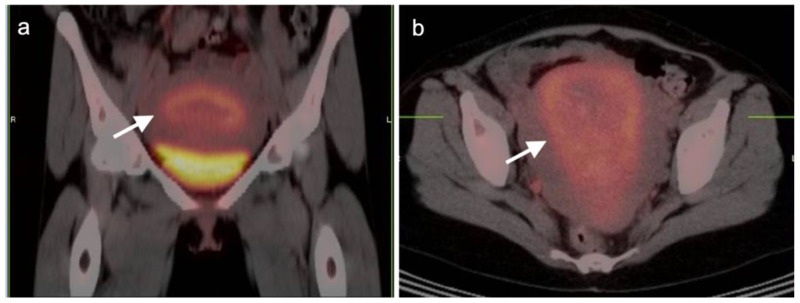
Post-partum uterus. (**a**) Coronal and (**b**) axial FDG-CT images of the pelvis show an enlarged uterus with a thickened rind of metabolically active endometrium (white arrow). Note that it is the endometrium, not the myometrium, that is hypermetabolic.

**Figure 4 cancers-14-03000-f004:**
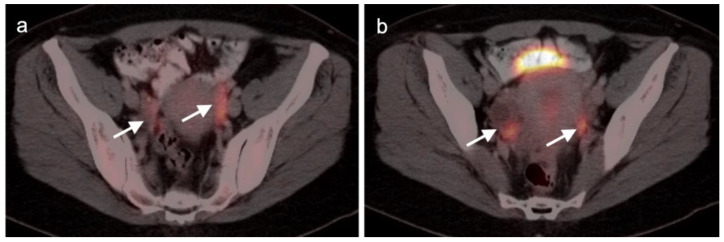
Physiologic [^18^F]F-fluoro-2-deoxy-d-glucose (FDG) uptake in fallopian tubes. (**a**,**b**) Axial FDG-CT images of the pelvis show increased metabolic activity in the fallopian tubes (white arrows). This typically occurs during the mid-cycle and during ovulation and can be unilateral or bilateral.

**Figure 5 cancers-14-03000-f005:**
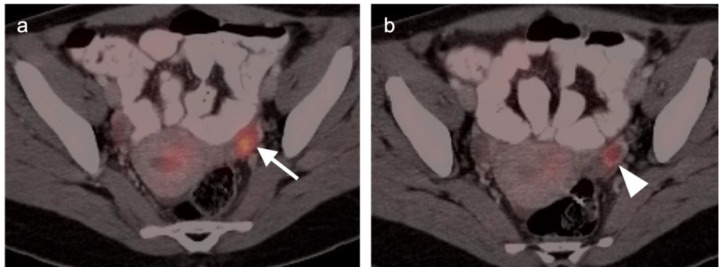
Normal functional ovarian metabolic activity on axial FDG PET-CT images of the pelvis. (**a**) Maturing ovarium follicle with a homogeneous nodular focus of increased metabolic activity (white arrow); (**b**) Corpus luteum cyst with a ring enhancing, hypermetabolic nodule (white arrowhead).

**Figure 6 cancers-14-03000-f006:**
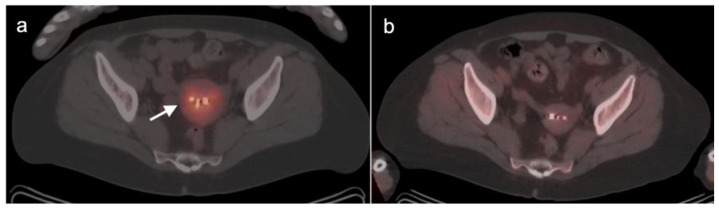
Pelvic inflammatory disease (PID shown on axial FDG PET-CT images of the pelvis. (**a**) For a few weeks following placement of an intrauterine contraceptive device (IUD), patients are at increased risk of developing pelvic inflammatory disease (PID), shown here as increased metabolic activity in the endometrium and myometrium on FDG PET-CT (white arrow). (**b**) A later FDG PET-CT scan shows that the PID and metabolic activity have resolved.

**Figure 7 cancers-14-03000-f007:**
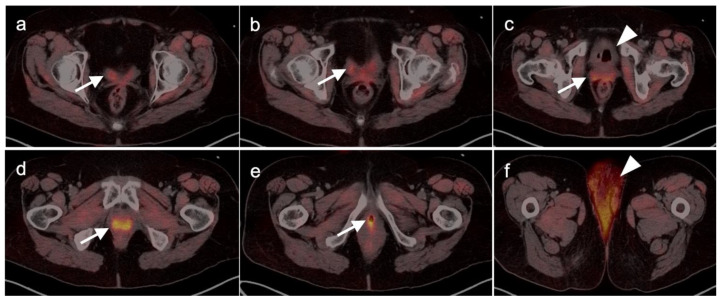
Vesicovaginal fistula on axial FDG PET_CT images of the pelvis. The patient complained of severe incontinence to urine. A vesicovaginal fistula resulted from prior pelvic radiation. Findings on sequential axial FDG PET-CT image of the pelvis include: (**a**–**e**) The vagina is diffusely filled with radioactive urine (white arrows); (**c**) The urinary bladder is empty and contains air (white arrowhead); (**f**) Radioactive urine is soaking the sanitary pad (white arrowhead).

**Figure 8 cancers-14-03000-f008:**
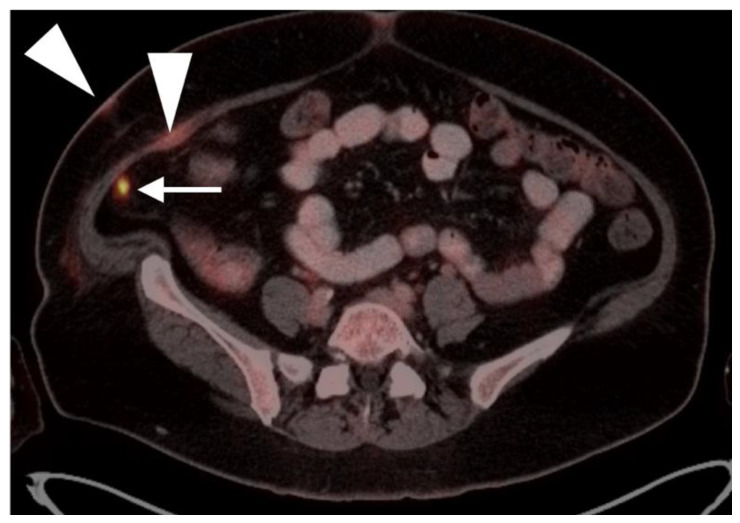
Seeding of the peritoneum with ovarian cancer with laparoscopic removal. Laparoscopic removal of ovarian cancer carries a higher risk of seeding the peritoneum with tumor, seen here on an axial FDG PET-CT image of the abdomen, with a small hypermetabolic tumor nodule (white arrowhead) near a laparoscopic port site (white arrowhead).

**Figure 9 cancers-14-03000-f009:**
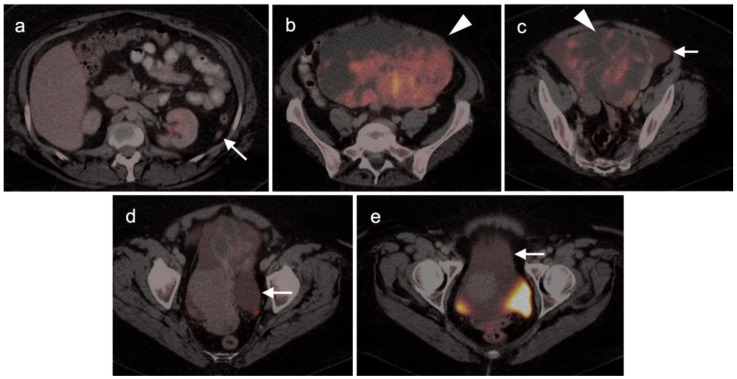
Mucinous cystadenocarcinoma of the ovary with peritoneal spread. (**a**–**e**) Axial FDG PET-CT images of the abdomen. (**a**) Small peritoneal implant (white arrow); (**b**,**c**) large mixed attenuation ovarian mass (white arrowheads). The solid components are hypermetabolic. (**c**–**e**) Areas of subtly metabolically active mucinous ascites (white arrows).

**Figure 10 cancers-14-03000-f010:**
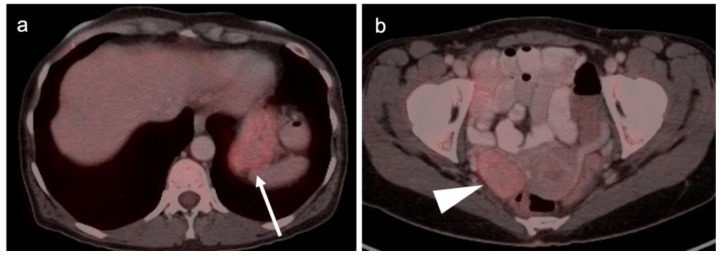
Gastric carcinoma with a Krukenberg tumor of the ovary shown on axial FDG PET-CT images of the abdomen and pelvis. (**a**) Axial FDG PET-CT images of the upper abdomen show the stomach to be only mildly metabolically active in this patient with a signet-ring adenocarcinoma of the stomach (white arrow). It would be difficult to distinguish this tumor on PET-CT from normal gastric activity. (**b**) Axial FDG PET-CT of the pelvis reveals a mildly metabolically active right ovarian mass (white arrowhead) which was a Krukenberg tumor (metastatic from gastric cancer). Krukenberg tumors account for 1–2% of ovarian tumors.

**Figure 11 cancers-14-03000-f011:**
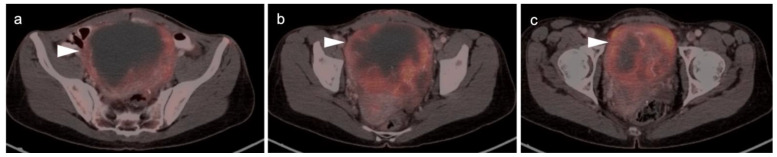
Mucinous adenocarcinoma of the colon metastatic to the ovary shown on axial FDG PET-CT images of the pelvis. (**a**–**c**) Axial FDG PET-CT images of the pelvis. There is a hypermetabolic, centrally cystic tumor in the pelvis (white arrowheads), which at surgery was a large metastasis to the ovary from a colon cancer. The ovaries represent a common site for metastatic disease for many tumors, including from colon cancer. This mucinous adenocarcinoma metastasis to the ovary is very similar in appearance on FDG PET-CT to primary ovarian mucinous carcinomas.

**Figure 12 cancers-14-03000-f012:**
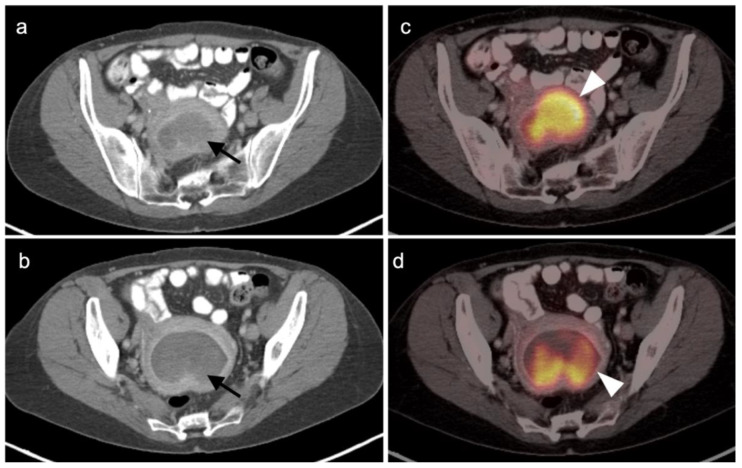
Endometrial carcinoma show on axial FDG PET-CT images of the pelvis. (**a**,**b**) Axial contrast-enhanced CT images of the pelvis show an endometrial mass that is very low in attenuation (black arrows). (**c**,**d**) Although low in attenuation, the mass is intensely hypermetabolic on FDG PET-CT (white arrowheads) and is typical in appearance for endometrial carcinoma.

**Figure 13 cancers-14-03000-f013:**
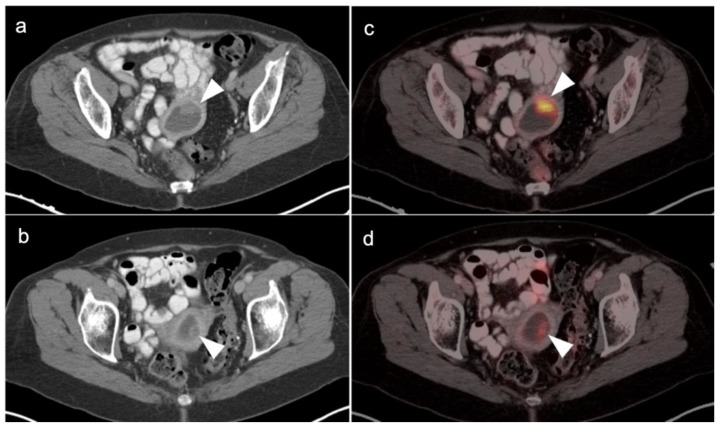
Endometrial polyp shown by axial FDG-PET_CT images of the pelvis in a 54-year-old woman with a long history of tamoxifen use for breast cancer. (**a**,**b**) Axial contrast enhanced CT of the pelvis shows a fluid-filled endometrial cavity containing a low-attenuating polypoid mass (white arrowheads). (**c**,**d**) Axial FDG PET-CT shows that the low attenuating polypoid lesion is intensely hypermetabolic (white arrow), a histologically proven to be benign hyperplasic polyp. A small endometrial carcinoma would have a similar appearance.

**Figure 14 cancers-14-03000-f014:**
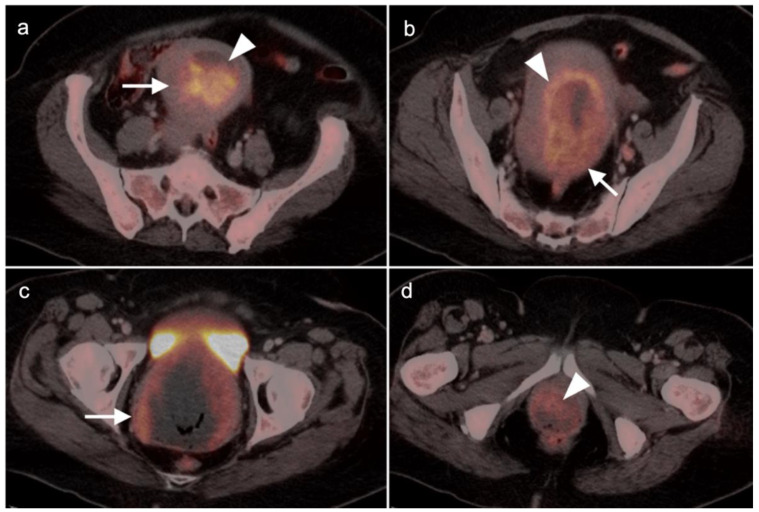
Endometrial stromal sarcoma on axial FDG PET-CT images of the pelvis. (**a**,**b**) A hypermetabolic endometrial mass fills the endometrial cavity (white arrowheads) and is centrally necrotic. (**c**,**d**) There are multiple areas of myometrial invasion (white arrows). Histologically, this was a high-grade endometrial stromal sarcoma.

**Figure 15 cancers-14-03000-f015:**
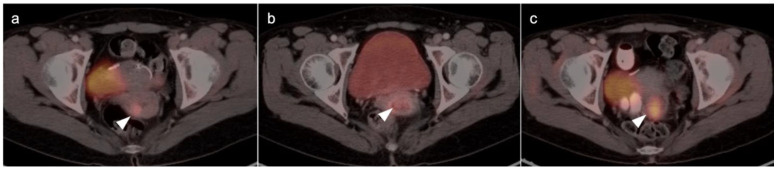
Stage IB1 cervical cancer shown on axial FDG PET-CT images of the pelvis. (**a**) Small cervical cancer (1.8 cm diameter, white arrowhead); (**b**) post radiation with persistent metabolic activity in the endocervix consistent with post-treatment inflammation vs. persistent tumor (white arrowhead); (**c**) 2 years later, cervical cancer recurred with a larger tumor (white arrowhead).

**Figure 16 cancers-14-03000-f016:**
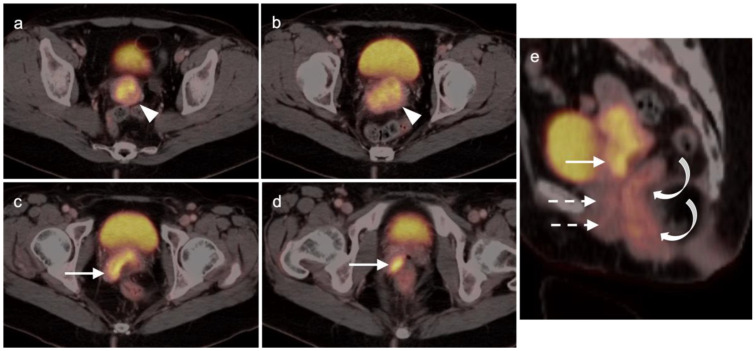
Stage IIA cervical cancer shown by FDG PET-CT images of the pelvis. (**a**,**b**) Axial FDF PET-CT of the pelvis shows a hypermetabolic tumor involving the entire cervix (white arrowhead); (**c**,**d**) Tumor extends into the vagina on the right (white arrowhead). (**e**) Sagittal view shows cervical cancer with extension into the upper third of the vagina (white arrow), a normal appearing lower two-thirds of the vagina (dashed white arrows) and normal rectum (curved white arrows).

**Figure 17 cancers-14-03000-f017:**
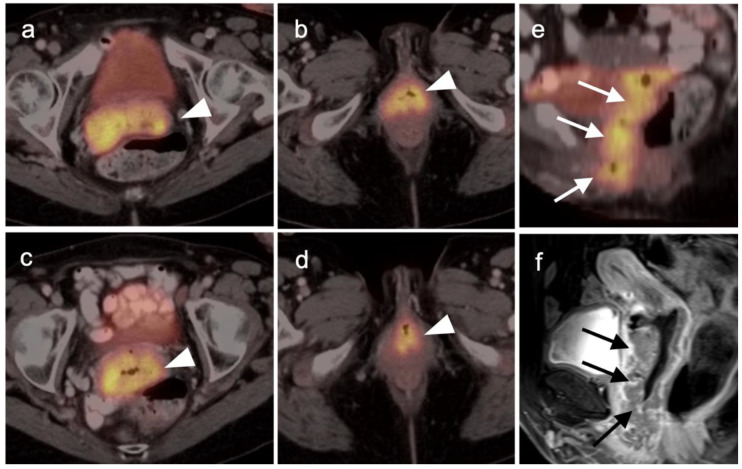
Stage IIIA cervical cancer by FDG PET-CT and MRI. (**a**–**d**) Axial FDG PET-CT demonstrates extension of cervical cancer throughout the entire vagina (white arrowheads). (**b**,**e**) Sagittal views of the pelvis shows cervical cancer spread throughout the entire vagina by FDG PET-CT ((**e**), white arrows) and MRI ((**f**), black arrows).

**Figure 18 cancers-14-03000-f018:**
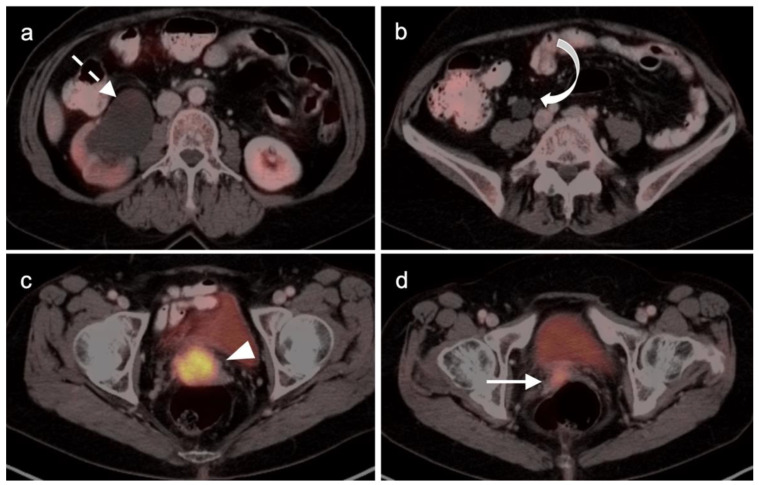
Stage IIIB cervical cancer by axial FDG PET-CT images of the pelvis (**a**) Hydronephrosis (white dashed arrow); (**b**) hydroureter (white curved arrow); (**c**) hypermetabolic cervical cancer (white arrowhead; (**d**) involvement of the right vaginal apex near the expected location of the right ureterovesicle junction (white arrow).

**Figure 19 cancers-14-03000-f019:**
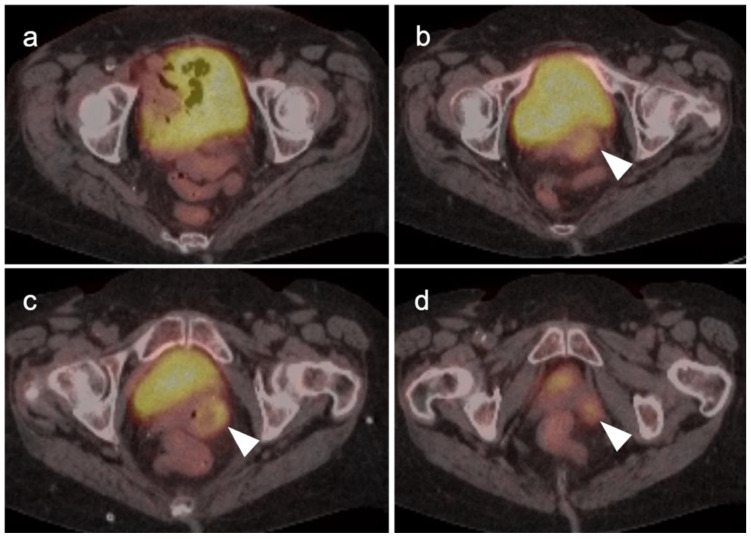
Primary vaginal squamous cell carcinoma on axial FDG PET-CT images of the pelvis. (**a**) shows normal upper pelvic structures; (**b**–**d**) moderately hypermetabolic mass in the left vaginal apex (white arrowheads).

**Figure 20 cancers-14-03000-f020:**
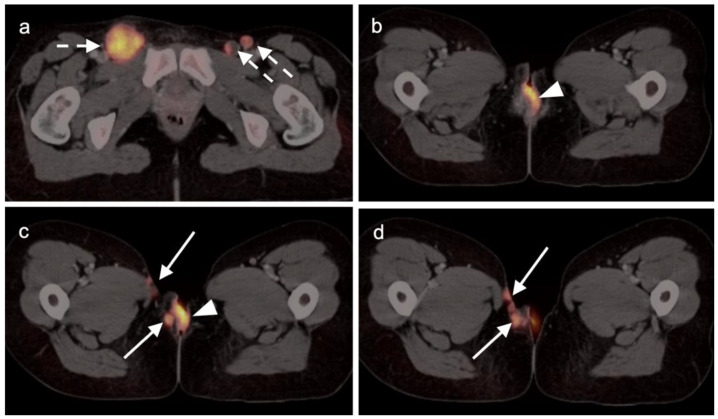
Vulvar squamous cell carcinoma on axial FDG PET-CT images of the pelvis. (**a**) Axial FDG PET-CT demonstrates bilateral inguinal lymph node involvement (white dashed arrows); (**b**,**c**) the vulvar carcinoma is markedly hypermetabolic on FDG PET-CT (white arrowhead); (**c**,**d**) there is spread of the vulvar carcinoma into the adjacent subcutaneous tissue on the right, as well as the skin of the right medial thigh (white arrows).

**Figure 21 cancers-14-03000-f021:**
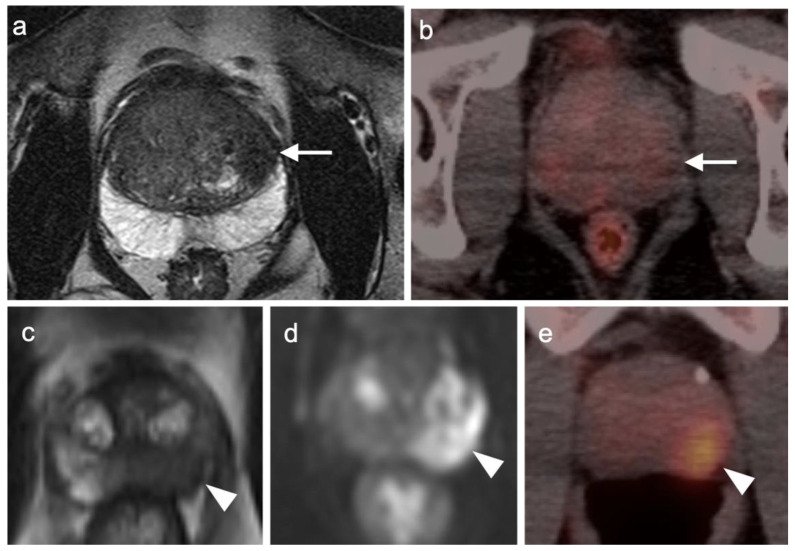
Two patients with prostate cancer on axial MRI and FDG PET-CT images of the pelvis. Case 1: (**a**) Well-differentiated prostate cancer (Grade group 2) shows typical low signal on T2 MRI in the peripheral zone of the left lobe of the prostate (white arrow); (**b**) FDG PET-CT in this patient is negative in the region of tumor (white arrow) but shows mild diffuse uptake typical for benign prostatic hypertrophy (BPH); Case 2: (**c**) T2 MRI in a patient with more poorly differentiated prostate cancer (Grade group 4) shows typical low signal intensity in a tumor in the peripheral zone of the left lobe of the prostate (white arrow); (**d**) DWI MRI shows increased signal intensity in the prostate cancer (white arrowhead); (**e**) FDG PET-CT shows increased metabolic activity in the prostate cancer (white arrowhead).

**Figure 22 cancers-14-03000-f022:**
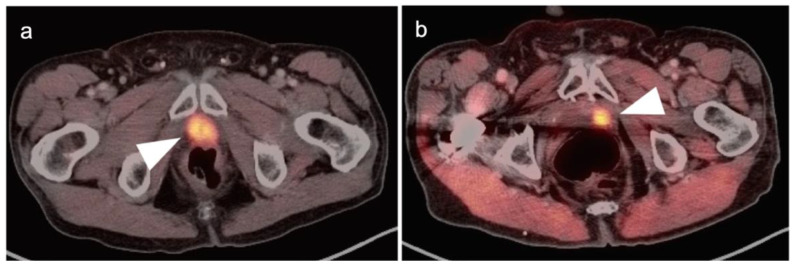
Two patients with prostate cancer recurrence in the post-surgical prostatic fossa, clearly delineated by [^18^F]-fluciclovine PET-CT. (**a**,**b**) The prostate cancer recurrence is intensely PET-positive in both cases (white arrowheads).

**Figure 23 cancers-14-03000-f023:**
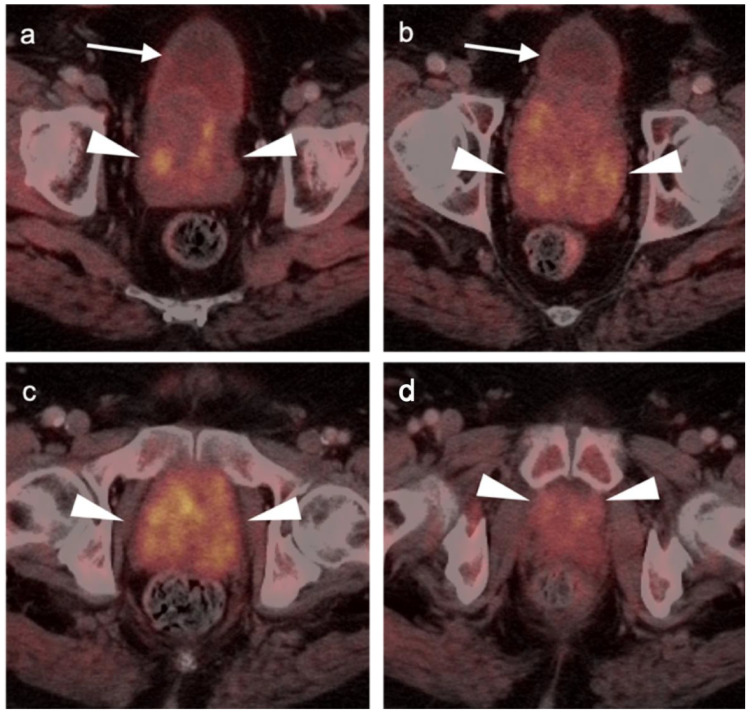
Benign prostatic hypertrophy on axial [^18^F]-fluciclovine PET-CT images of the pelvis. (**a**–**d**) The prostate is diffusely enlarged and heterogeneously hypermetabolic (white arrowheads). The detection of a site of prostate cancer within this prostate would be difficult. (**a**,**b**) White arrows identify the urinary bladder.

**Figure 24 cancers-14-03000-f024:**
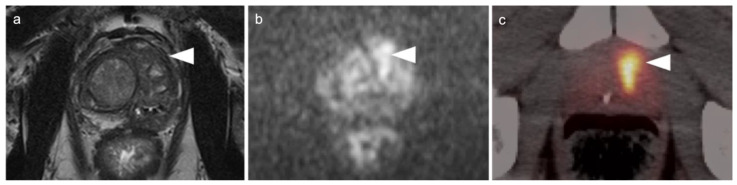
Prostate cancer in a patient with benign prostatic hypertrophy (BPH), compared by multiparametric (mp) MRI (**a**,**b**) and [^18^F]-DCFPyL PET-CT (**c**). (**a**) T2 MRI fails to clearly delineate the prostate cancer (site shown by white arrowhead) with the background of benign prostatic hypertrophy (PBH) nodules; (**b**) DWI MRI shows high signal intensity due to prostate cancer (white arrowhead); (**c**) the prostate cancer is best shown on the [^18^F]-DCFPyL PET-CT (white arrowhead). The background of BPH is low in activity on [^18^F]-DCFPyL PET-CT, which offers an advantage over [^18^F]-fluciclovine PET-CT, which typically shows diffuse heterogeneous uptake of tracer in BPH.

**Figure 25 cancers-14-03000-f025:**
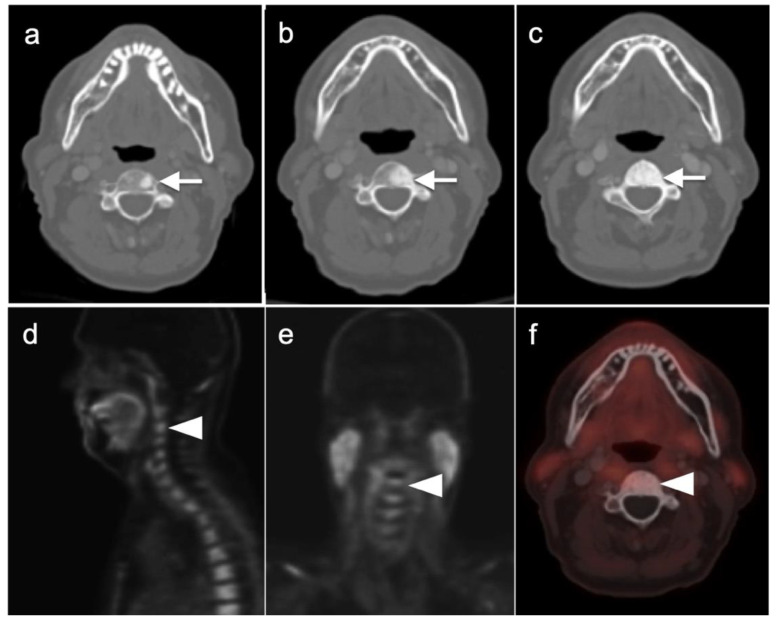
**[**^18^F]-Fluciclovine-negative PET-CT scans in a patient with metastatic bone disease to an upper cervical vertebra. (**a**–**c**) There is progressive sclerosis of the vertebra in 3 successive CT scans performed with **[**^18^F]-Fluciclovine PET-CT scans over a two-year interval (left to right); (**d**–**f**) the last ^18^F-fluciclovine PET-CT scan shows no increased uptake in the bone metastasis over that of normal background marrow.

**Figure 26 cancers-14-03000-f026:**
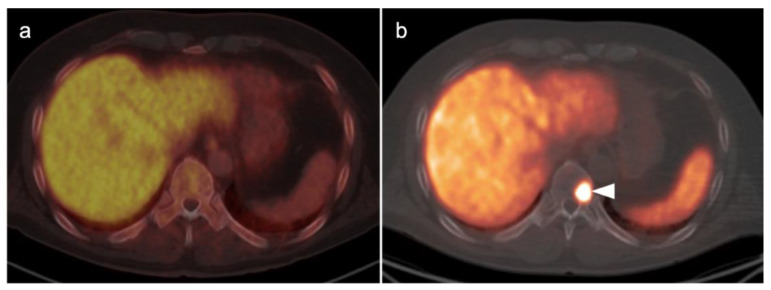
Comparison of [^18^F]-fluciclovine negative but [^18^F]-PSMA-positive PET-CT scans in a prostate cancer patient with a bone metastasis. (**a**) [^18^F]-fluciclovine PET-CT scan is negative; (**b**) [^18^F]-PSMA scan is positive in a metastatic lesion of a lower thoracic vertebral body (white arrowhead) vertebral metastasis.

**Figure 27 cancers-14-03000-f027:**
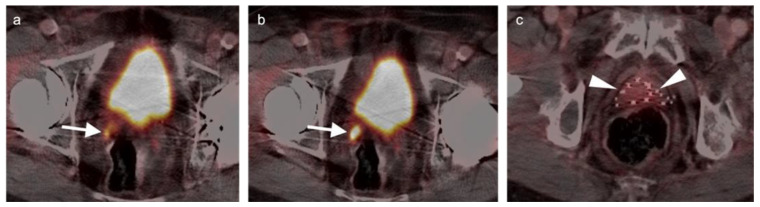
Diagnostic value of PSMA PET-CT detecting extra-prostatic recurrence in the face of metal artifact. (**a**,**b**) There is prostate cancer recurrence in the right seminal vesicle (white arrows), identified despite the presence of bilateral total hip prostheses. A metal artifact reduction algorithm was utilized for the CT; (**c**) Multiple brachytherapy seeds are present (white arrowheads) without obvious recurrence within the prostate.

**Figure 28 cancers-14-03000-f028:**
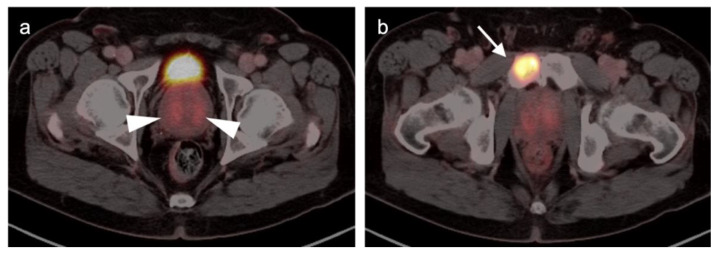
Discordant uptake between primary tumor and metastatic prostate cancer on [^18^F]-DCFPyL (PSMA) PET-CT. (**a**) The primary tumor was Grade Group 3 in the transitional zone of both lobes of the prostate, corresponding to the regions of mild uptake on PMSA PET-CT (white arrowheads). There is less activity on PMSA PET-CT than would normally be expected for unknown reasons. The patient had not received ADT; (**b**) a metastasis to the right pubic bone (white arrow) is intensely PET positive, in comparison to the primary tumor.

**Figure 29 cancers-14-03000-f029:**
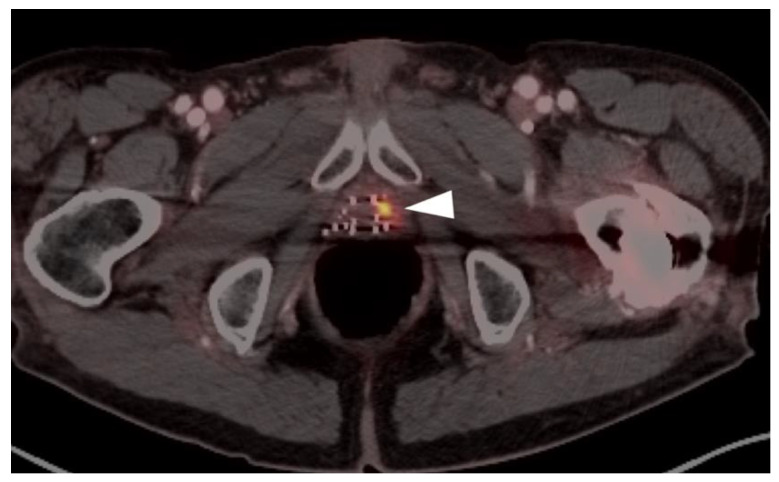
A patient with prior brachytherapy with recurrent prostate cancer in the left lobe of the prostate, shown on PSMA PET-CT. The site of tumor is very small but clearly delineated (white arrowhead). MRI would have had limited utility in this patient because of metal artifact created by the brachytherapy seeds.

**Figure 30 cancers-14-03000-f030:**
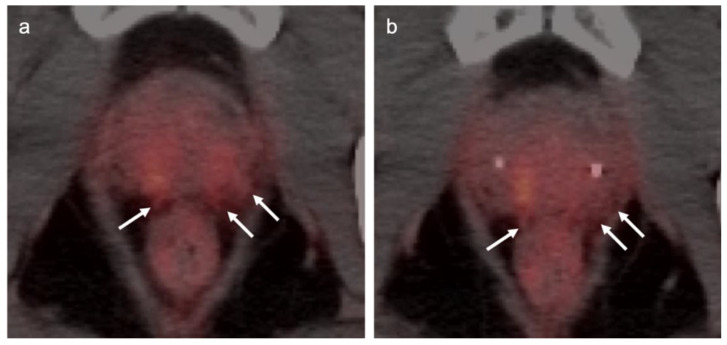
Subtle extra-prostatic spread of prostate cancer on PSMA PET-CT. (**a**,**b**) Axial PSMA PET-CT images demonstrate moderately PET-positive tumor in the posterior peripheral and transitional zones of both lobes of the prostate. The posterior capsule of the prostate is indistinct and shows stranding soft tissue that is PET avid (white arrows), supporting extracapsular spread of disease. The patient had bilateral transitional and peripheral zone tumor by biopsy.

**Figure 31 cancers-14-03000-f031:**
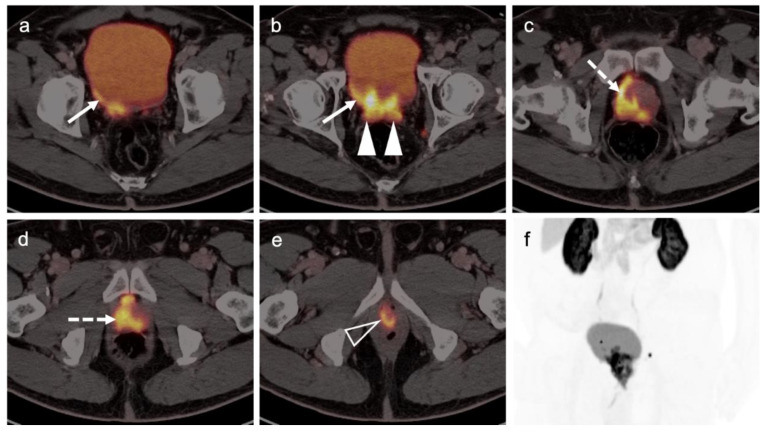
Extensive extra-prostatic spread of prostate cancer on PSMA PET-CT. Areas of tumor include (**a**,**b**) spread into the urinary bladder (white arrows); (**b**) involvement of both seminal vesicles (solid white arrowheads); (**c**,**d**) extensive involvement of the prostate; (**e**) encasement of the proximal urethra (open white arrowhead); (**f**) MIP PSMA PET mage of the pelvis shows the extensive extra-prostatic involvement and two small involved bilateral pelvic lymph nodes.

**Figure 32 cancers-14-03000-f032:**
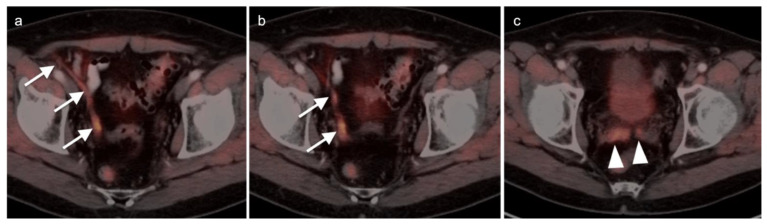
Gross extra-prostatic spread of prostate cancer to vas deferens and seminal vesicles on [^18^F]-Fluciclovine PET-CT. (**a**,**b**) There is PET-positive tumor extending along the right vas deferens (white arrows); (**c**) tumor also involves both seminal vesicles (white arrowheads).

**Figure 33 cancers-14-03000-f033:**
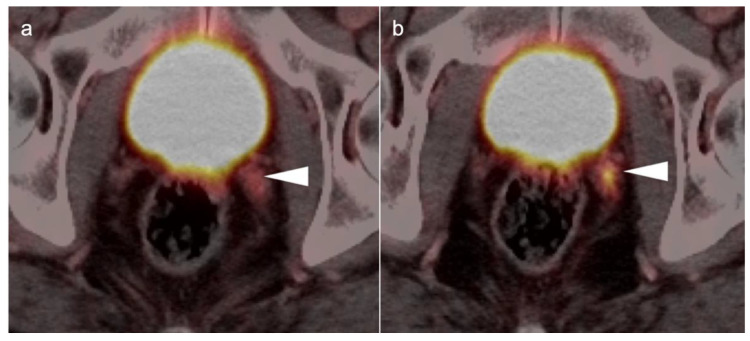
Tumor involving the left neurovascular pedicle by PSMA PET-CT (**a**,**b**). This patient had a radical prostatectomy and seminal vesicles were removed. Involvement of the left neurovascular pedicle (white arrowhead) could be confused with seminal vesicle without knowledge of the surgery performed.

**Figure 34 cancers-14-03000-f034:**
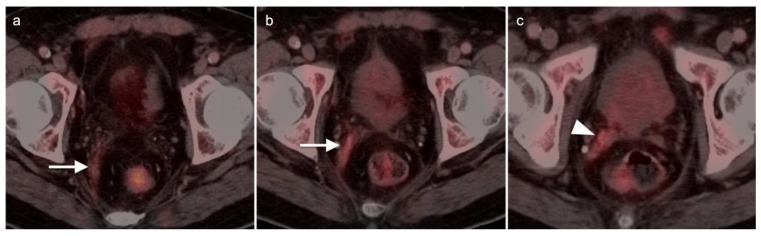
[^18^F]-fluciclovine PET-CT in a post prostatectomy patient showing recurrent tumor in the right neurovascular pedicle ((**c**), white arrowhead), extending along the right mesorectal fascia ((**a**,**b**), white arrows), representing a path of perineural spread along the hypogastric plexus.

**Figure 35 cancers-14-03000-f035:**
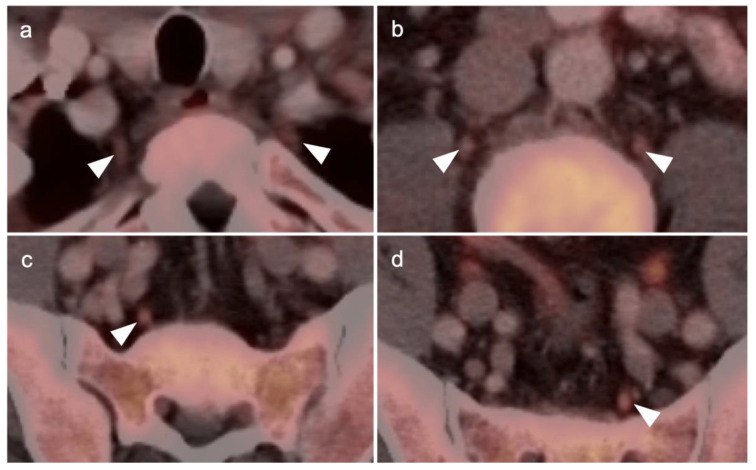
[^18^F]-Fluciclovine PET-CT showing mild uptake in the sympathetic ganglia (white arrowheads). (**a**) Stellate ganglia; (**b**) lumbar ganglia; (**c**,**d**) sacral ganglia. PSMA PET-CT shows a similar pattern of uptake in the sympathetic ganglia. Celiac ganglia are similarly seen (not shown here).

**Figure 36 cancers-14-03000-f036:**
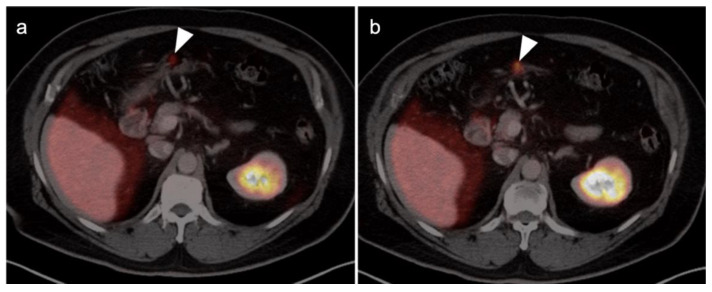
[^18^F]-PSMA PET-CT demonstrates tiny focus of intense activity in the wall of the stomach (**a**,**b**, white arrows), shown by subsequent endoscopic biopsy to be a small GIST tumor.

**Figure 37 cancers-14-03000-f037:**
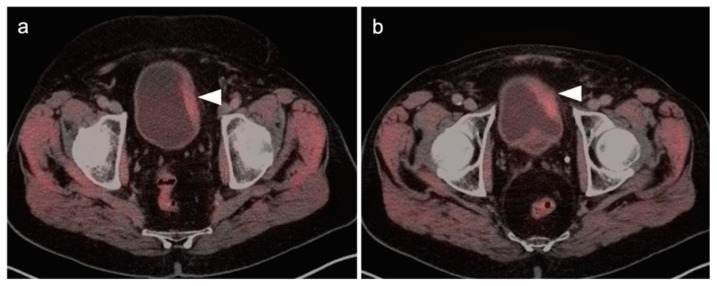
[^18^F]-Fluciclovine PET-CT demonstrates PET-positive left-sided bladder wall thickening (**a**,**b**, white arrowheads), biopsy-proven urothelial carcinoma.

**Figure 38 cancers-14-03000-f038:**
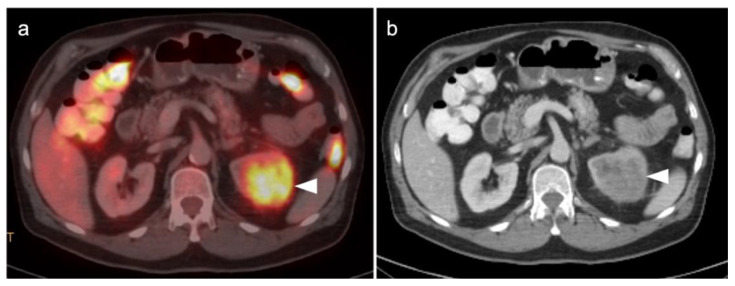
Upper track urothelial carcinoma shown on FDG PET-CT images of the pelvis. (**a**) Intensely hypermetabolic mass on an axial FDG PET-CT image of the upper abdomen involving the left kidney (white arrowhead); (**b**) corresponding contrast-enhanced CT scan shows a soft tissue mass filling the left renal pelvis with parenchymal invasion (white arrowhead).

**Figure 39 cancers-14-03000-f039:**
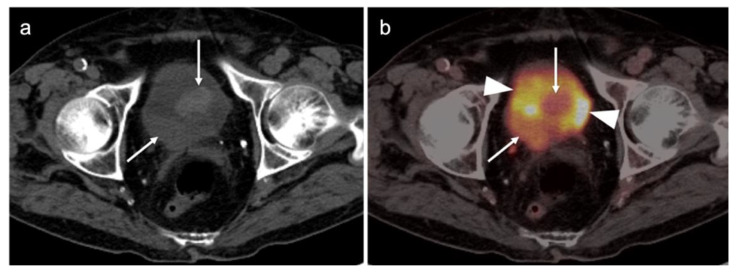
Lower tract urothelial carcinoma. (**a**) Axial non-contrast enhanced CT image shows a mass arising from the right posterior aspect of the urinary bladder with a lobular component extending into the bladder (white arrows); (**b**) Axial FDG PET-CT image shows that the mass in the bladder is hypermetabolic (white arrows), but lower in activity than the surrounding urine (white arrowheads).

**Figure 40 cancers-14-03000-f040:**
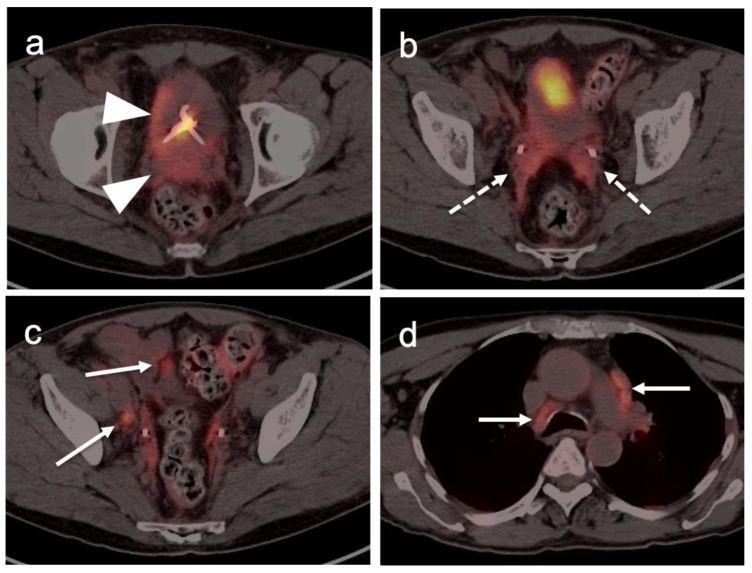
Lower tract urothelial cancer with metastatic disease shown on axial FDG PET-CT images. (**a**) The primary tumor is hypermetabolic (white arrowhead). (**b**) There is diffuse infiltrative spread into the seminal vesicles and mesorectal fascia (dashed white arrows). (**c**) There are multiple hypermetabolic nodes in the pelvis and mesentery (white arrows). (**d**) There are multiple hypermetabolic mediastinal lymph nodes (biopsy proven metastatic, white arrows).

**Figure 41 cancers-14-03000-f041:**
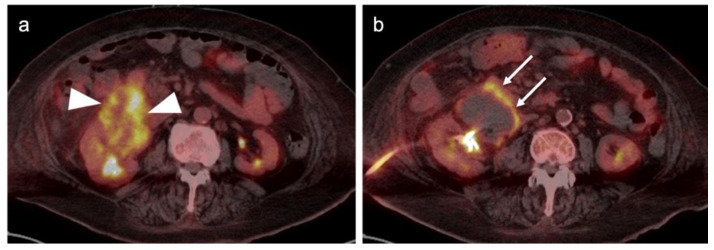
Upper tract urothelial carcinoma pre- and post chemotherapy on FDG PET-CT. (**a**) There is a large hypermetabolic renal pelvic mass prior to treatment (white arrowheads). (**b**) Following chemotherapy, there is a persistent mass that is centrally necrotic but shows a peripheral rim of metabolic activity suggesting persistent viable tumor (white arrows). There is a percutaneous nephrostomy tube.

**Figure 42 cancers-14-03000-f042:**
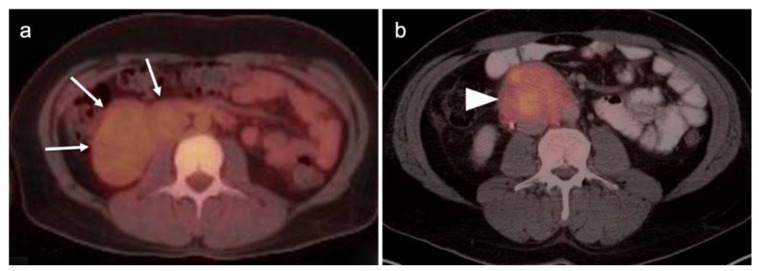
Two cases of large pure seminoma demonstrating variability in metabolic activity in retroperitoneal masses on FDG PET-CT. (**a**) An axial FDG PET-CT image shows a large tumor with relatively mild uptake (white arrow), relative to background muscle and other tissues; (**b**) Another large retroperitoneal site of seminoma with more intense metabolic activity is shown on FDG PET-CT (white arrowhead).

**Figure 43 cancers-14-03000-f043:**
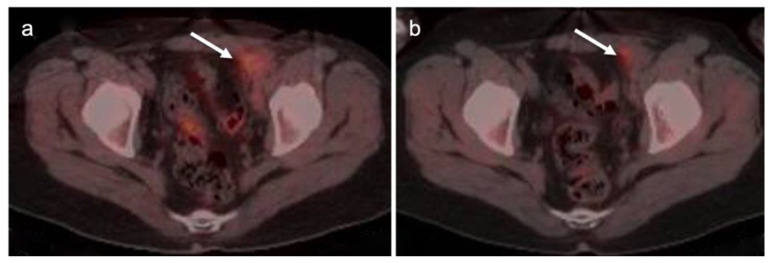
Pre- (**a**) and post-treatment (**b**) FDG PET-CT scans in a mixed germ cell tumor (white arrows) show a reduction in size following treatment but residual metabolic activity in the left inguinal mass. Subsequent stability in size over a long interval supported that no residual viable tissue was present post-treatment, despite residual metabolic activity.

**Figure 44 cancers-14-03000-f044:**
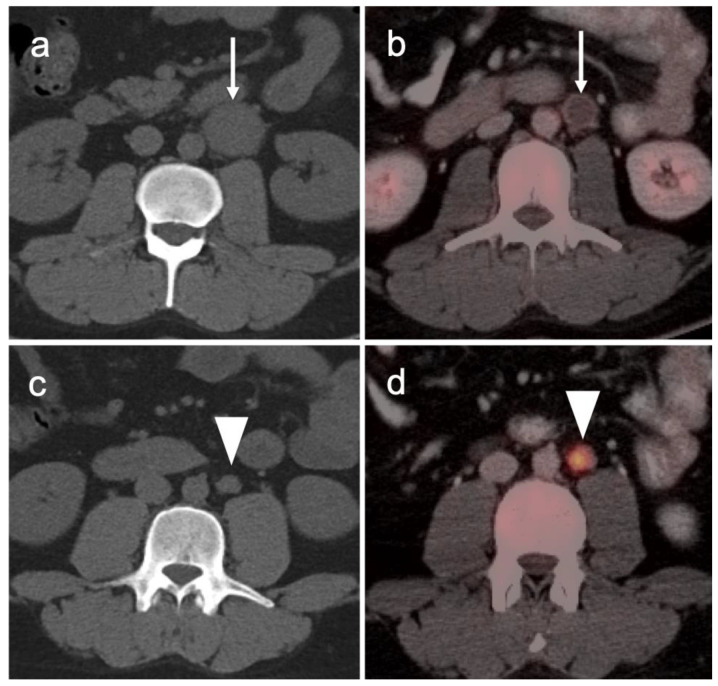
A pure seminoma with retroperitoneal metastases showing a variable response to treatment is shown on a pre-treatment CT (left, **a**,**c**) and a post-treatment FDG PET-CT (right, **b**,**d**). Pretreatment CT (**a**) shows a sizable retroperitoneal mass (white arrow) that undergoes a decrease in size and attenuation, as well as an absence of metabolic activity on post treatment FDG PET-CT (**b**), white arrow); However, a more inferior additional left periaortic nodule (**c**, white arrowhead) on the pre-treatment CT shows an interval increase in size as well as significant metabolic on the post-treatment FDG PET-CT (**d**, white arrowhead), consistent with tumor progression.

**Figure 45 cancers-14-03000-f045:**
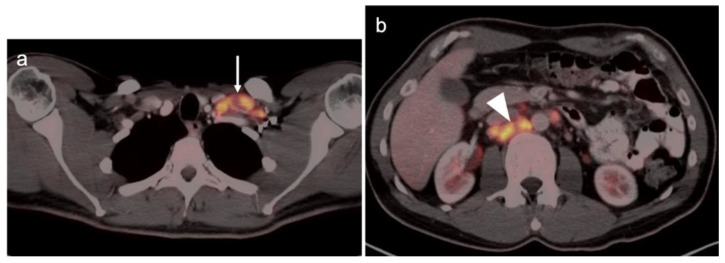
Intensely hypermetabolic metastatic embryonal germ cell tumor of the testis on FDG PET-CT to lymph nodes in the left supraclavicular region (**a**, white arrow) and the retroperitoneum (**b**, white arrow).

**Figure 46 cancers-14-03000-f046:**
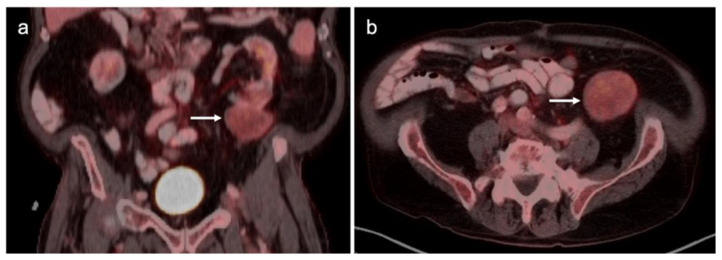
Renal cell carcinoma. FDG PET-CT coronal (**a**) and axial (**b**) images. An exophytic mass (white arrows) arising from the inferior pole of the left kidney is slightly higher in metabolic activity than normal renal parenchyma. This is typical for renal cell carcinoma, which can also be similar in metabolic activity to the renal parenchyma.

**Figure 47 cancers-14-03000-f047:**
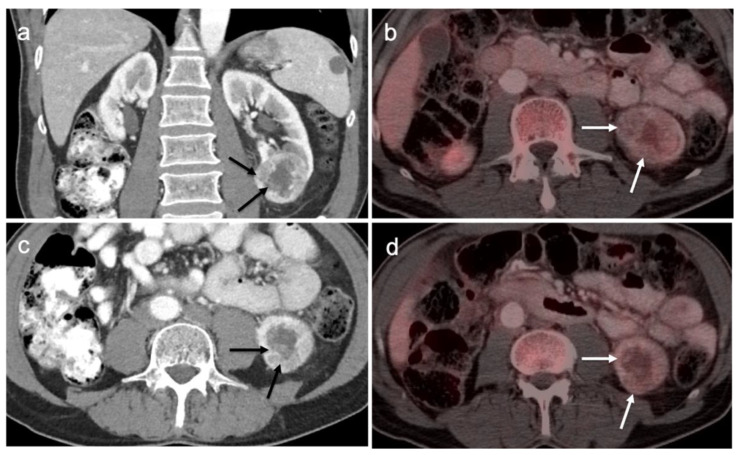
Renal oncocytoma on contrast enhanced FDG PET-CT. (**a**,**c**) Contrast-enhanced CT shows a heterogeneously enhancing mass (black arrows) in the left kidney; (**b**,**d**) On FDG PET-CT, the oncocytoma would be difficult to appreciate without the concurrent contrast enhanced CT and is similar in metabolic activity to the remainder of the normal renal parenchyma (white arrows). This was an oncocytoma, but a renal cell carcinoma can have a similar in appearance.

**Figure 48 cancers-14-03000-f048:**
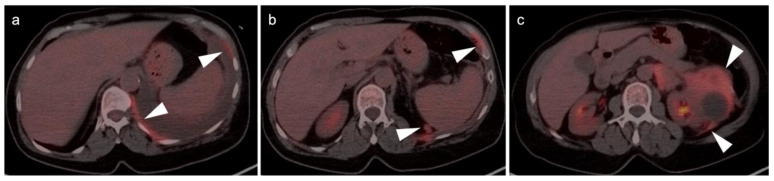
Sarcomatoid cell carcinoma on FDG PET-CT. (**c**) A large, hypermetabolic invasive mass arising from the left kidney extends into the pararenal space (white arrowhead); (**a**,**b**) The hypermetabolic renal mass extends into the left pleural space, with an accompanying pleural effusion (white arrowheads). This was a sarcomatoid renal cell carcinoma, but the main differential would also include lymphoma.

**Figure 49 cancers-14-03000-f049:**
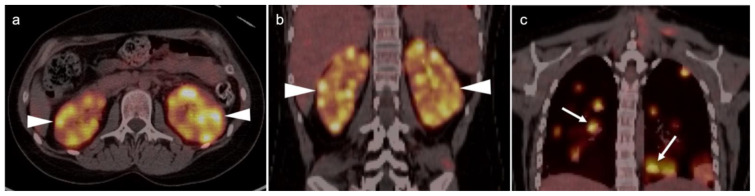
Renal lymphoma on FDG PET-CT. (**a**,**b**) Numerous hypermetabolic renal nodules are present in both kidneys, as shown on axial (**a**) and coronal (**b**) FDG PET-CT images (white arrowheads); (**c**) Multiple hypermetabolic nodules are present within the lungs (also lymphoma), as shown on coronal FDG PET-CT image (white arrows). This was diffuse large B-cell lymphoma with renal and lung involvement.

**Figure 50 cancers-14-03000-f050:**
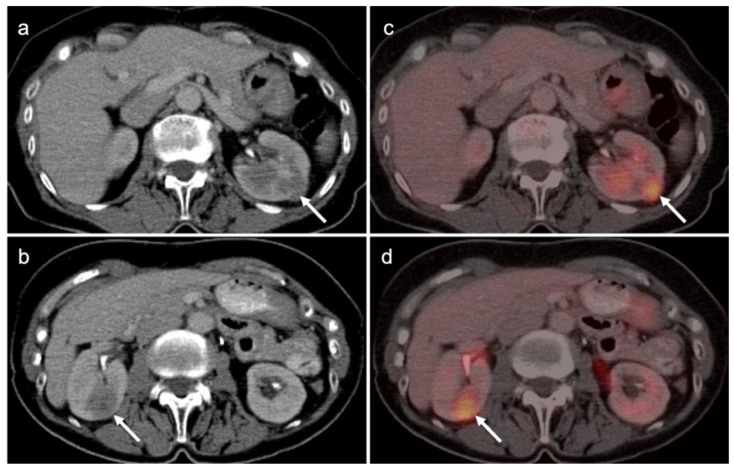
Metastases to the kidneys. (**a**,**b**) Contrast-enhanced axial CT images of the abdomen show multiple hypoattenuating soft tissue nodules in both kidneys (white arrows); (**c**,**d**) on corresponding axial FDG PET-CT images, the hypoattenuating lesions are hypermetabolic. These lesions were metastatic from an ovarian primary malignancy.

**Figure 51 cancers-14-03000-f051:**
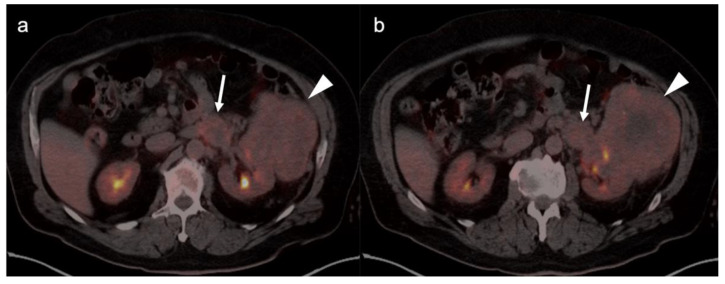
Large renal cell carcinoma with renal vein invasion. FDG PET-CT (**a**,**b**) shows a large left renal mass extending into the pararenal space (white arrowhead) and invading the left renal vein (white arrow).

**Figure 52 cancers-14-03000-f052:**
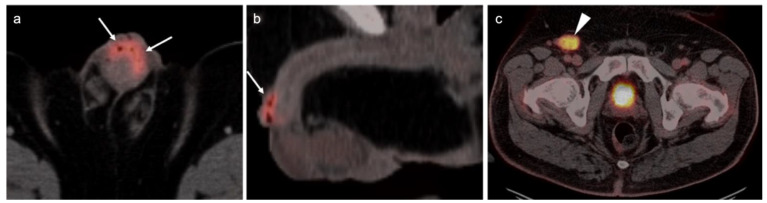
Squamous cell carcinoma of the prepuce of the penis on FDG PET-CT. Axial (**a**) and sagittal (**b**) FDG PET-CT images of the penis demonstrate intensely hypermetabolic tissue at the dorsal penile prepuce (white arrows). (**c**) Axial FDG PET-CT of the pelvis shows involvement of an enlarged hypermetabolic right inguinal lymph node (white arrowhead).

**Figure 53 cancers-14-03000-f053:**
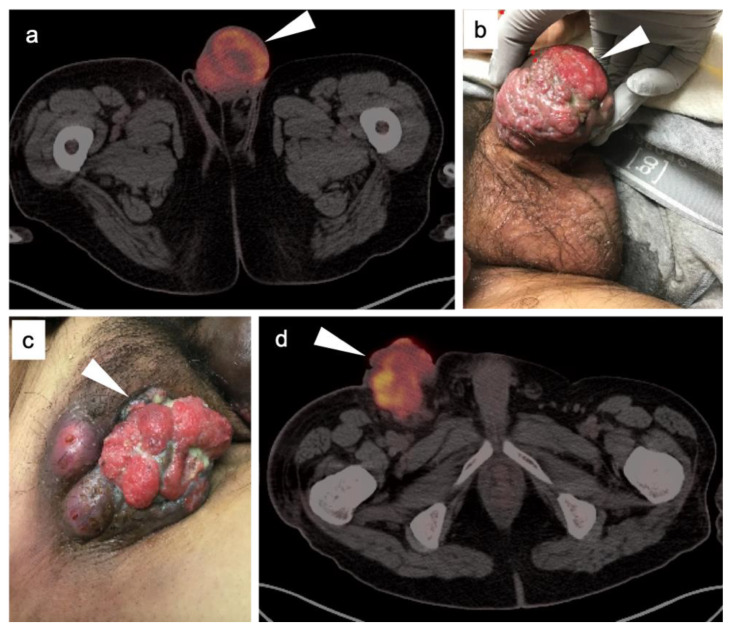
Large squamous cell carcinoma of the penis with fungating right inguinal lymph node shown on FDG PET-CT. (**a**) Axial FDG PET-CT image shows very large penile hypermetabolic mass (white arrowhead). (**b**) Photograph of the tumor shows large lobular, erythematous mass involving the glans penis (white arrowhead) extending into the penile shaft. (**c**) Photograph of the right inguinal lymph node shows a large, fungating, ulcerated right inguinal nodal mass that has extended through the skin (white arrowhead). (**d**) Axial FDG PET-CT of the right inguinal lymph node mass shows a large, lobular hypermetabolic mass in the right inguinal region.

**Figure 54 cancers-14-03000-f054:**
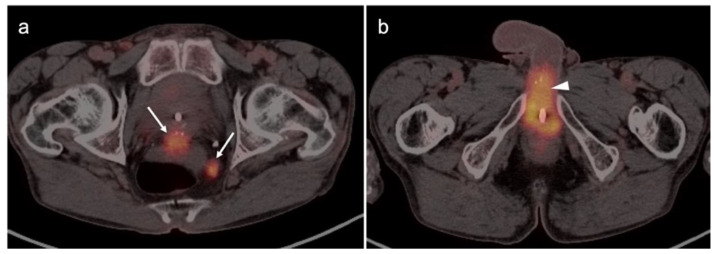
Squamous cell carcinoma of the base of the penis, spread from a cutaneous perineal squamous cell carcinoma. (**a**) Axial FDG PET-CT at left shows intrapelvic tumor deposits (white arrows). (**b**) Hypermetabolic tumor involves the base of the penis (white arrowhead). A Foley catheter is in place.

**Figure 55 cancers-14-03000-f055:**
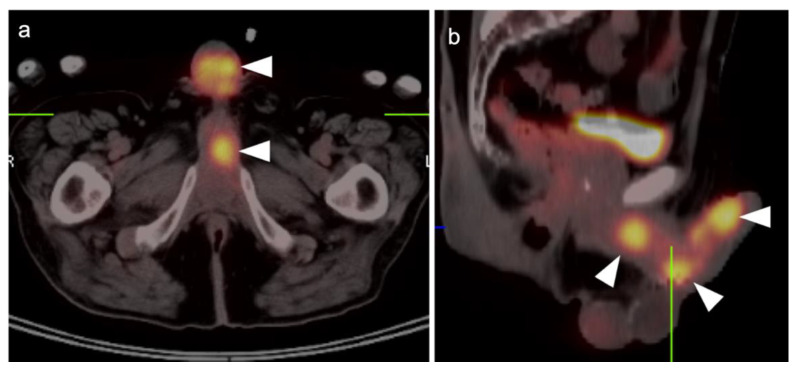
Metastases to the penis. Axial (**a**) and sagittal (**b**) FDG PET-CT images demonstrate multiple hypermetabolic masses involving the penis (white arrowheads). The primary tumor was a sinonasal undifferentiated carcinoma (SNUC).

**Table 1 cancers-14-03000-t001:** NCCN classification of unfavorable intermediate to very high-risk prostate cancer (appropriate for initial staging by PSMA PET-CT) *.

Risk Group	Criteria
Unfavorable Intermediate risk	Grade Group 3 and cT2b, cT2c or PSA 10–20 ng.mL No high risk features >50% biopsy cores (+)
High risk	Has at least one of the following: cT3a Grade Group 4 or Grade Group 5 PSA > 20 ng/mL
Very high risk	Has at least one of the following: cT3b–T4 Primary Gleason pattern 5 2–3 high risk features >4 cores with Grade Group 4 or 5

* Adapted from NCCN guidelines 3.2022 [84].

**Table 2 cancers-14-03000-t002:** New histological grade group classification of prostate cancer *.

Grade Group	Gleason Score	Gleason Pattern
Grade 1	≤6	≤3 + 3
Grade 2	7	3 + 4
Grade 3	7	4 + 3
Grade 4	8	4 + 4, 3 + 5, 5 + 3
Grade 5	9–10	4 + 5, 5 + 4, 5 + 5

* Adapted from NCCN guidelines 3.2022 [84].

**Table 3 cancers-14-03000-t003:** Prostate (intact). Interpretive criteria for ^18^F fluciclovine (Axumin^®^) for the prostate in patients with an intact prostate post definitive treatment. This includes patients who have had brachytherapy, external beam radiotherapy or other ablative therapy, such as cryotherapy *.

Prostate versus Reference Region	Interpretation	Note
Uptake between blood pool and bone marrow	Equivocal	Poor specificity, tissue confirmation recommended
Diffuse, focal or multifocal uptake > bone marrow	Suspicious for malignancy	Poor specificity, tissue confirmation recommended
<1 cm focus, uptake > blood pool	Suspicious for malignancy	Poor specificity, tissue confirmation recommended
<blood pool	Likely benign	data

* Adapted from *Eur. J. Nucl. Med. Mol.* Imaging 2020 [87].

**Table 4 cancers-14-03000-t004:** Seminal vesicles. Interpretive criteria for ^18^F fluciclovine (Axumin^®^) for the seminal vesicles in patients with prostate cancer *.

Seminal Vesicles versus Reference Region	Interpretation	Note
Symmetrical, <blood pool	Most likely physiologic	Seminal vesicles are typically removed with prostatectomy, so suspicious uptake in this region may be in the neurovascular pedicle
Focal uptake or >blood pool	May be malignant	Recommend MRI

* Adapted from *Eur. J. Nucl. Med. Mol.* Imaging 2020 [87].

**Table 5 cancers-14-03000-t005:** Prostatectomy bed (fossa). Interpretive criteria for ^18^F fluciclovine (Axumin^®^) for the prostatic fossa in patients post prostatectomy *.

Fossa versus Reference Region	Interpretation	Note
Uptake > bone marrow	Suspicious for malignancy	Urine activity is often > blood pool Urethra often has prominent activity Pubococcygeus and anal sphincter muscles show uptake
Uptake > blood pool if focus < 1 cm	Suspicious for malignancy	
Uptake between blood pool and bone marrow	Does not meet criteria for malignancy but requires close follow-up	
<blood pool	Most likely benign	

* Adapted from *Eur. J. Nucl. Med. Mol.* Imaging 2020 [87].

**Table 6 cancers-14-03000-t006:** Lymph nodes. Interpretive criteria for ^18^F fluciclovine (Axumin^®^) for lymph nodes in patients with prostate cancer *.

Lymph Node Region	Uptake Relative to Reference Region	Interpretation	Additional Notes
Typical location	>1 cm long axis >bone marrow	Equivocal	Typical location is from pelvis to renal vessels Inflammatory nodes false + Symmetrical distal external iliac or inguinal usually false + Stellate, sacral, lumbar, and celiac sympathetic ganglia show activity
Typical location	>1 cm long axis between blood pool and bone marrow	Does not meet criteria for malignancy but requires close follow-up	
Typical location	<1 cm long axis, >blood pool and approaching bone marrow	Suspicious for malignancy	
Atypical location	Mild symmetric activity	Likely benign	
Atypical location	Asymmetric activity (>bone marrow)	Suspicious for malignancy	

* Adapted from *Eur. J. Nucl. Med. Mol.* Imaging 2020 [87].

**Table 7 cancers-14-03000-t007:** PSMA-RADS structured reporting system for PSMA-targeted PET-CT for prostate cancer (PCa) *.

PSMA-RADS Number	Interpretation	Description
PSMA-RADS 1A	Benign	Benign lesion by CT and PET
PSMA-RADS 1B	Benign	Lesion characterized by biopsy or addition imaging to be benign
PSMA-RADS 2	Likely benign	Low level uptake in site not typical for prostate cancer metastasis, or uptake in bone lesion more likely to be alternative diagnosis.
PSMA-RADS 3A	Equivocal	Borderline soft tissue site of uptake soft tissue, typical location for PCa involvement
PSMA-RADS 3B	Equivocal	Equivocal bone lesion
PSMA-RADS 3C	Equivocal	Marked uptake at soft tissue or osseous site atypical of prostate cancer
PSMA-RADS 3D	Equivocal	Lesion suspicious on anatomic imaging but lacking uptake on PET
PSMA-RADS 4	PCa highly likely	Marked uptake but lacking confirmation on anatomic imaging (includes small nodes)
PSMA-RADS 5	PCa almost certainly present	Marked uptake with anatomic imaging confirmatory

* Adapted from *J. Nucl. Med.* 2018 [89].

**Table 8 cancers-14-03000-t008:** miPSMA expression scoring system, equating likelihood of malignancy to ratios of activity in lesions to that in normal reference tissues *.

Score	Reported PSMA Expression	Uptake in Lesion Relative to that in Normal Reference Tissues
0	None	Uptake < blood pool
1	Low	>Blood pool but <liver
2	Intermediate	>liver but <parotid
3	High	>parotid

* Adapted from *J. Nucl. Med.* 2017 [90]. Can be used for both ^68^Ga PSMA-11 and ^18^F DCFPyL.

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
