# Peer review of "PET-CT in Clinical Adult Oncology—IV. Gynecologic and Genitourinary Malignancies"

_cancers, 2022, doi:10.3390/cancers14123000_

Round 1
Reviewer 1 Report
I reviewed with great interest the Review entitled "PET-CT in Clinical Adult Oncology – IV. Gynecologic and 2 Genitourinary Malignancies" submitted by Salem and co-authors.
The review is part of a series of 6 reviews on PET/CT in oncology.
- Please define the materials and methods and the design of the review. It is not clear the method adopted for the selection of papers. Did the authors use QUADAS and PRISMA as methods for article selection?
- In view of the recent development of AI and Radiomics based analysis in several studies, I suggest including for each disease any development or the current status of AI on PET imaging.
- One of the added values of PET imaging is the predictive role on survival outcomes and its impact on clinical management. Please discuss more accurately these aspects for each disease.
- Different articles have not been listed /cited.
Eg. PMID:
31468240
26268680
32814979
35406542
35608445
Reviewer 2 Report
I have no comments, it is a very good paper which should be published as it is.Author Response
See attached

Reviewer 3 Report
Excellent review. Valuable for any physician. I haven comments
Reviewer 4 Report
Ahmed Ebada Salem reviewed interesting topics regarding PET-CT in Clinical Adult Oncology.
Points to be considered:
1) The rationale of why the authors came up with this review.
2) What is the information that is not exactly available that motivated the authors to come up with this information. What are the current caveats and how do the authors highlight the current research in answering them? If not they need to address in future directions.
3) IAs is now well known, tumors grow and evolve through a constant crosstalk with the surrounding microenvironment, and emerging evidence indicates that angiogenesis and immunosuppression frequently occur simultaneously in response to this crosstalk (can the author expand and highlight the teragnostic role for imaging)?.
4) Accordingly, strategies combining anti-angiogenic therapy and immunotherapy seem to have the potential to tip the balance of the tumor microenvironment and improve treatment response: please refer to PMID: 32456352 and expand in light of their findings
5) The authors need to come up with a graphical abstract representing the key findings and a workflow
6) The authors need to highlight what new information the review is providing to enhance the research in progress.
Round 2
Reviewer 1 Report
The article has been improved following the reviewer's comments.
Reviewer 4 Report
The authors have clarified several of the questions I raised in my previous review. Most of the major problems have been addressed by this revision.